



# Balloon-borne Stratospheric Vertical Profiling of Carbonyl Sulfide and Evaluation of Ozone Scrubbers

Alessandro Zanchetta[1], Steven van Heuven[1], Joram Hooghiem[1,6], Rigel Kivi[3], Thomas Laemmel[4], Michel Ramonet[4], Markus Leuenberger[5], Peter Nyfeler[5], Sophie L. Baartman[6], Maarten Krol[6,7], and Huilin Chen[1,2]

[1] Center for Isotope Research (CIO), Energy and Sustainability Research Institute Groningen (ESRIG), University of Groningen, Groningen, the Netherlands
[2] Joint International Research Laboratory of Atmospheric and Earth System Sciences, School of Atmospheric Sciences, Nanjing University, Nanjing, China
[3] Space and Earth Observation Centre, Finnish Meteorological Institute (FMI), Sodankylä, Finland
[4] Laboratoire des Sciences du Climat et l'Environnement (LSCE), CEA - CNRS - UVSQ - University Paris-Saclay, Gif sur Yvette, France
[5] Climate and environmental Physics, Physics Institute and Oeschger Centre for Climate Change Research, University of Bern, Bern, Switzerland
[6] Meteorology and Air Quality, Wageningen University & Research, Wageningen, the Netherlands
[7] Institute for Marine and Atmospheric Research, Utrecht University, Utrecht, the Netherlands

Corresponding to Huilin.Chen@nju.edu.cn or Huilin.Chen@rug.nl

**Abstract**

Carbonyl sulfide (COS) is a low abundant atmospheric trace gas that has a tropospheric lifetime of 2-2.5 years, allowing it to reach the stratosphere, where it undergoes photolysis and reactions with OH• and O• radicals, generating precursors of stratospheric aerosols. Vertical profiling of COS has rarely been realised, especially for stratospheric observations. In this study, we introduce a new technique for continuous and discrete vertical profiling of COS based on the analysis of air samples collected by AirCore, the LIghtweight Stratospheric Air (LISA) sampler and its scaled-up version BigLISA in three campaigns in Trainou (2019), Kiruna (2021) and Sodankylä (2023) using a Quantum Cascade Laser Spectrometer (QCLS). To eliminate potential COS measurement biases, we have investigated the efficiency of different scrubbers based on cotton and squalene for removing ozone ($O_3$) and assessed their potential impacts on COS measurement. Furthermore, we examined the influence of different inlet configurations and $O_3$ scrubbers on the retrieved COS profiles, and found no significant impact within the uncertainties. We found that the differences with the averaged profiles obtained from the Atmospheric Chemistry Experiment – Fourier Transform Spectrometer (ACE-FTS) and the measured AirCore profiles at both mid and polar latitudes were less than 5%, and approximately 10% for the LISA samples at polar latitudes. Differences between our observations and COS observations from the *SPectromètre InfraRouge d'Absorption à Lasers Embarqués* (SPIRALE) ranged from 10% to 15%, with





both methods showing similar COS trends over altitude. Moreover, we found squalene-based scrubbers to be suitable for quantitative $O_3$ removal. Both AirCore and the LISA samplers are lightweight and suitable for routine balloon-borne COS profiling, providing useful observations for stratospheric research and validation of COS retrievals from remote sensing techniques.

## 1. Introduction


Carbonyl sulfide (COS, also referred at as OCS) is an odorless and colorless gas species (Ferm, 1957). It is the most abundant sulfur-containing gas species in the atmosphere, with a tropospheric mole fraction of 350-500 parts per trillion (ppt, pmol/mol) (Berry et al., 2013; Remaud et al., 2023). It has been suggested as a proxy to partition photosynthetic uptake of $CO_2$ from respiration, to improve the quantification of carbon fluxes between atmosphere and vegetation (Campbell et al., 2008; Montzka

et al., 2007; Sandoval-Soto et al., 2005; Stimler et al., 2009; Whelan et al., 2018). Given its relatively long tropospheric lifetime of 2-2.5 years (Ma et al., 2021; Montzka et al., 2007; Remaud et al., 2023), COS can reach the stratosphere, where it is converted to sulfur dioxide ($SO_2$), a precursor of stratospheric aerosols, by photolysis and reactions with OH• and O• radicals (Brühl et al., 2012; Chin and Davis, 1995; Krysztofiak et al., 2015). Althought the debate has not been fully resolved, COS is considered to likely be the largest contributor to stratospheric sulfur aerosols during volcanic quiescent periods (Brühl et al.,

2012; Crutzen, 1976; Kremser et al., 2016).

Currently, observations of stratospheric COS vertical profiles and/or total columns are performed by ground- and satellite-based remote sensing (Barkley et al., 2008; Bernath, 2005; Toon et al., 2018) and by deploying balloon-borne spectrometers (Krysztofiak et al., 2015; Toon et al., 2018). The collection of air samples for COS analysis was only carried out sporadically

or only in the upper troposphere/lowermost stratosphere (10-12 km) (Engel and Schmidt, 1994; Karu et al., 2023). In this paper, we present new techniques to collect continuous and discrete stratospheric air samples, based on the balloon-borne AirCore (Karion et al., 2010) and the LIghtweight Stratospheric Air (LISA) (Hooghiem et al., 2018) and BigLISA samplers, respectively, paired with a Quantum-Cascade Laser Spectrometer (QCLS, Aerodyne Research Inc., MA, USA, model TILDAS-CS) for COS measurements. These methods allow analysis of collected air samples with minimal preparation and

treatment, which reduces risks of contamination during sampling and storage.

Possible impacts of stratospheric ozone ($O_3$) (Engel and Schmidt, 1994) as well as pollution-induced tropospheric $O_3$ (Andreae et al., 1990, 1993; Hofmann et al., 1992; Persson and Leck, 1994) on collected air samples for COS observations have been reported in previous studies. Since stratospheric $O_3$ is more abundant than pollution-induced tropospheric $O_3$ (in particular

between 15 and 35 km of altitude, where $O_3$ mixing ratios may reach roughly 10 ppm) (World Meteorological Organization (WMO), 1999), its impact on air samples for COS observations may be significant. Therefore, we have investigated different techniques to remove $O_3$ before sampling, and assessed their potential impacts on the mole fractions of COS and other trace



gases. In particular, we have investigated different $O_3$ scrubbers and their scrubbing efficacy and effect on COS, also by deploying different inlets on the aforementioned samplers. Furthermore, we show a comparison of measured continuous and

discrete COS samples with previous observations. A particular focus is set on a cross-validation comparison with SPIRALE's in situ spectrometry (Krysztofiak et al., 2015) and ACE-FTS remote sensing COS observations (Bernath, 2005; Glatthor et al., 2017; Velazco et al., 2011).

## 2. Materials and methods

### 2.1 Samplers

The three different samplers deployed to collect stratospheric air will be presented in this section. All instruments flew under weather balloons that typically reached altitudes of 30 to 35 km. The presented data was collected in different campaigns, namely the RINGO campaign in Trainou (TRN, France, 2019), the HEMERA campaign (Schuck et al., 2025) in Kiruna (KRN, Sweden, 2021) and the OSTRICH campaign at the Fourier Transform Spectrometer (FTS) site (Kivi and Heikkinen, 2016) in Sodankylä (SOD, Finland, 2023). An overview of campaigns and samplers is reported in Table 1. Air samples collected by all

devices were analysed on a QCLS, which will be described in Sect. 2.2.

### 2.1.1 AirCore

The AirCore sampler was first introduced by Karion et al. (2010) to retrieve $CO_2$ and $CH_4$ vertical profiles. It consists of a long, stainless-steel tube usually shaped as a coil, internally coated with Sulfinert® to prevent reactions or adsorption of gas species with the tube walls (Karion et al., 2010; Membrive et al., 2017). When used to retrieve vertical profiles, AirCore

sampling is realised passively (Karion et al., 2010; Membrive et al., 2017; Wagenhäuser et al., 2021). Before the flight, the coil is filled with a known gas mixture, which will later help identifying the starting point of the AirCore profiles during analysis. During ascent, the coil empties through one open end due to decreasing ambient pressure. After the balloon bursts, the instrument collects air during descent while ambient pressure is increasing, without using a pump. Knowing sampling pressure and temperature, and assuming pressure equilibrium during the filling process, each aliquot of moles of air can be

calculated for each sampling altitude interval. These can be associated to the aliquot of analysed moles of air in the coil, allowing the retrieval of vertical profiles (Karion et al., 2010; Membrive et al., 2017; Tans, 2022). However, the selected start and end point of each analysis, air mixing inside the coil, sample loss, and general fill dynamics may all be causes for deviations from this approximation (Membrive et al., 2017; Tans, 2022; Wagenhäuser et al., 2021). Detailed discussions of fill dynamics and uncertainties treatment can be found in Tans (2022) and Membrive et al. (2017), respectively. In this study, we followed

the approach described by Membrive et al. (2017) to retrieve the altitude mapping and the relative uncertainties along the vertical profiles, with one exception, as follows.

As reported in Table 1, flight SOD3 included a double-sided sampling AirCore, property of the University of Bern, which was flown and analysed by our group. One half of this 200 m long AirCore was equipped with an oxygen-spiking system,





programmed to inject 5 minuscule shots of $O_2$ in the coil as altitude markers at 21045, 17005, 11837, 7870 and 4592 m.
Although our QCLS is not capable of measuring $O_2$, these injections were visible as COS anomalies along the profile (see
Sect. 4.1). Therefore, the altitude mapping for this AirCore was realised by matching the COS spikes with the reported spiking
altitudes. Moreover, the cotton scrubber installed on one of the two inlets likely adsorbed water ($H_2O$) before the ascent phase,
which was then taken in the AirCore at the beginning of the descent, mixing tropospheric $H_2O$ with stratospheric air at the
highest altitude. This required a dilution correction, followed by a matrix effect correction inferred from the correlation of
different species with $H_2O$ mole fraction.

At the altitude ceiling of the balloon flight, the AirCore's coil still contains part of the fill gas. This remaining fill gas can mix
with the sampled air at the top of the profile. Similarly, air from the lowest part of the sampled profile mix with the gas used
to push the air out of the coil during analysis. Therefore, the highest and the lowest parts of the profiles are flagged and are not
used for further analysis.

For the double-sided AirCore deployed on flight SOD3, fill gas flagging starts at lower altitude compared to other AirCores.
Given the design of this AirCore, air is collected from both ends of the coil, allowing the simultaneous sampling of two profiles.
Therefore, the top of the profiles and the remaining fill gas are located at the centre of the coil and not at one of its ends. For
all other AirCores, the top of the profile is the first part to be analysed, while for this sampler it will have to travel 100 m
through the coil before reaching the analyser. Thus, during analysis the top of the profile and the remaining fill gas have a
longer time to get mixed compared to other AirCores. Moreover, the resulting gas mixture may also experience a stronger
smearing effect due to its path through the coil. Altogether, this determines a larger portion of the profile that cannot be
considered for analysis compared to other AirCores.

Some AirCores experienced COS contamination due to specific design features (e.g., differential pressure sensors along the
coil or glue connections). Consequently, the contaminated COS mole fractions have been removed and are shown as gaps. The
causes of contamination are discussed in Sect. 4.1.

### 2.1.2 LISA sampler

The LISA sampler used in this study is a further miniaturised (55 L package size) and light-weight (2.9 kg) version of the
original sampler developed by Hooghiem et al. (2018). The instrument is battery-powered and is controlled by a
microcontroller, which also logs GPS, pressures, temperatures, and general instrument status. Differently from AirCore, LISA
actively pumps ambient air into four different 2.5L multi-layer foil (MLF) bags (type 30228-U, Supelco Inc., USA) through a
custom-made manifold. The sampling is performed during the ascent phase of the flight, since the vertical speed is slower than
during descent and this allows for a higher vertical resolution of the vertical sampling. The valves of the MLF bags are opened
and closed by servos. The sampling pressure intervals for each bag are programmed before the flight and limited to an absolute





pressure of 280 hPa to prevent bag burst after sampling. This is necessary because the ambient pressure continues to decrease during the remainder of the ascent, reaching about 10 hPa at 30 km altitude. The sampling pressures and the derived sampling altitude intervals and sample volumes are reported in Table 2. The mid-points of sample collections are calculated considering

the dependency of pump performance on the ambient pressure and the bag filling status, as described by Hooghiem et al. (2018).

Unusually high COS mole fractions were measured in laboratory tests and in some collected samples, which we speculate being due to outgassing from polymers (Lee and Brimblecombe, 2016). Therefore, during the SOD campaign in 2023, we

performed pre-conditioning of the MLF bags differently from what Hooghiem et al. (2018) described. Before each flight, MLF bags were not only filled and vacuumed with purified $N_2$, but filled and vacuumed repeatedly with air from a cylinder of synthetic air mixed with low mole fractions of $CH_4$, $CO_2$ and CO, which was meant to simulate stratospheric air conditions. This was done to prevent outgassing of different gas species, and in particular COS, from the polymers composing the MLF bags. The gas mixture used to flush the bags was measured on the QCLS before filling the bags and when it was pumped out

from them, allowing a control of potential contaminations under stratospheric sampling conditions. After the LISA sampler was recovered and brought back to the laboratory in the field, LISA air samples were transferred to glass flasks and stored for later analyses of COS and other trace gas species. Here we present the analysis results of the air samples left in the sampling bags, when present, directly after the sample transfer from the MLF bags to glass flasks (these latter were not analysed on the QCLS). The leftover volume of one of these samples was insufficient for analysis (SOD3 – L4), while two others showed

unusually high mole fractions for several of the analysed gas species (SOD2 – L4, SOD5 – L3). These three samples were labelled as outliers and will not be presented in this work, and are not included in Table 2.

### 2.1.3 BigLISA

BigLISA is a larger-volume and functionally improved variant of the LISA sampler, with a mass of roughly 12 kg. On the HEMERA missions in KRN, two independently operating BigLISA samplers were flown in a single enclosure on a high-

payload balloon (Schuck et al., 2025). Each BigLISA unit consists of a central box containing electronics and pneumatic components, and 6 externally mounted 10 L MLF bags (type 30229-U, Supelco Inc., USA). The BigLISA pneumatic hardware consists of a two-stage pump, a flow-reversing valve system that allows purging, a manifold and 6 closable MLF bags, all powered, monitored and controlled from a single control board.

Two stage-pumping is attained by connecting the four heads of two double-headed pumps (model NMP830.1.2KPDC-B HP, KNF, Germany) in a three to one configuration. This attains a high flow rate and a high compression factor. Laboratory tests confirmed the potential to reach a compression ratio of 25 at low ambient pressures. However, potentially due to unfavourably dimensioned connecting tubing, the compression ratio obtained during flight ranged between 2 and 5 times over ambient pressure.




Operation and logging are provided by a custom-made Printed Circuit Board (PCB) running an ESP32 microcontroller (Adafruit Industries, model HUZZAH32). Temperature is monitored at the pump heads, battery packs, power converters and within the larger BigLISA outer housing. In each pack, before and between pumping operations, the two 3-way valves are actuated as necessary to maintain in-pack temperature above 5 °C, preventing loss of battery capacity and conceivable pump

stalling. Power for electronics, pumps and valves is provided by two packs (for redundancy) of eight Saft LSH14 Li-SOCl$_2$ cells.

Housing for the two BigLISA units was provided by a customised high strength but lightweight aluminium frame, that was able to withstand the high accelerations potentially experienced during parachute deployment. In this structure, the two BigLISA packs are mounted centrally, surrounded by the 6 + 6 MLF bags, that are individually suspended using tie-wraps on

concentric stainless-steel wires. Protection from wind and radiation is provided by 1 mm thick aluminium sheeting. GPS receivers and sampling inlet pumping lines are led out radially at the top of the package.

During ascent, starting at 120 hPa (~15 km), bags were (re-)evacuated by the two-stage pump. This procedure removes conceivable traces of tropospheric air from the bags, and tests for plumbing integrity. From 30 hPa (~25 km), the manifold was flushed with ambient air to clear it of residual tropospheric air and water vapour. Just prior to collecting a sample, the

respective bag would be repeatedly filled with a tiny amount of air and evacuated, to dilute away any residual air in it. Sampling took place during the descent, for as long as it took to reach 200 hPa of sample pressure, but never more than 1800 seconds (less for lower samples), and would stop when the next sample was due to be collected. The sampling altitude intervals are reported in Table 2. The mid-points of sampling altitudes were calculated similarly to Hooghiem et al. (2018), considering the decreased pump performance when compression become necessary to fill the bag. However, given that BigLISA sampled

during descent, this effect was counteracted by the increase in ambient pressure during sampling. Therefore, the mid-points of BigLISA fall more towards the average of the sampling interval than the ones of LISA. Similarly to LISA, samples labelled as outliers due to unusually high mole fractions of multiple tracers (KRN - BL7, BL10, BL11 and BL12) will not be presented in this text.

**Table 1: overview of the instruments deployed for COS sampling, the launch location, samplers' sizes and additional features, and the inlet features in different campaigns. The reported diameters (Ø) refer to the outer diameter of the tubing. The flight code helps identifying which instruments have been deployed on the same balloons, and is used to refer to these flights in the text.**

| Location | Flight code | Flight date and takeoff time (UTC) | Apogee | Instrument | Instrument features | Inlet features |
|---|---|---|---|---|---|---|
| | | | | | | |





| | | | | | | |
|---|---|---|---|---|---|---|
| Trainou (FRA) 47°58' N 2°06' E | TRN1 | 17/06/2019 12:30 | 34.6 km | AirCore | 23 m x Ø 8 mm + 46 m x Ø 4 mm V ~ 1600 cm$^3$ | Free inlet |
| | TRN2 | 18/06/2019 8:00 | 32.6 km | AirCore | 36 m x Ø 3/16" + 38 m x Ø 1/8" V ~ 830 cm$^3$ | Mg(ClO$_4$)$_2$ dryer |
| | TRN3 | 18/06/2019 10:39 | 34.9 km | AirCore | 23 m x Ø 8 mm + 46 m x Ø 4 mm V ~ 1600 cm$^3$ | Free inlet |
| | TRN4 | 20/06/2019 07:07 | 32.3 km | AirCore | 36 m x Ø 3/16" + 38 m x Ø 1/8" V ~ 830 cm$^3$ | Free inlet |
| Kiruna (SWE) 67°53' N 21°04' E | KRN | 12/08/2021 21:18 | 33.1 km | AirCore | 37 m x Ø 3/16" + 39 m x Ø 1/8" V ~ 860cm$^3$ | Mg(ClO$_4$)$_2$ dryer |
| | KRN | 12/08/2021 21:18 | 33.1 km | AirCore | 36 m x Ø 3/16" + 38 m x Ø 1/8" V ~ 830 cm$^3$ | Free inlet |
| | KRN | 12/08/2021 21:18 | 33.1 km | BigLISA | 12 x 10 L MLF-bags | Mg(ClO$_4$)$_2$ dryer |
| Sodankylä (FIN) 67°22' N 26°37' E | SOD1 | 02/08/2023 7:00 | 29.7 km | AirCore | 40 m x Ø 1/4" + 58 m x Ø 1/8" V ~ 1400 cm$^3$ Differential pressure sensors | Mg(ClO$_4$)$_2$ dryer |
| | SOD2 | 02/08/2023 14:50 | 30.1 km | LISA | 4 x 2.5 L MLF-bags, pre-conditioned | Mg(ClO$_4$)$_2$ dryer |
| | SOD3 | 05/08/2023 10:56 | 26.3 km | LISA | 4 x 2.5 L MLF-bags, pre-conditioned | Free inlet |



| | | | | | |
|---|---|---|---|---|---|
| SOD3 | 05/08/2023 10:56 | 26.3 km | AirCore | 200 m x Ø 1.5 mm" V ~ 1400 cm³ Double-sided sampling | One side free inlet, one side cotton-based $O_3$ scrubber |
| SOD4 | 06/08/2023 9:03 | 29.8 km | LISA | 4 x 2.5 L MLF-bags, pre-conditioned | Cotton- based $O_3$ scrubber |
| SOD5 | 08/08/2023 6:17 | 29.3 km | AirCore | 40 m x Ø 1/4" + 58 m x Ø 1/8" V ~ 1400 cm³ Differential pressure sensors | $Mg(ClO_4)_2$ dryer and cotton-based $O_3$ scrubber |

**Table 2: pressure, altitudes and estimated sampled volumes at standard temperature and pressure (STP) conditions of the BigLISA (BL) and LISA (L) samples.**

| Flight code and sample code | Minimum ambient p (hPa) | Maximum ambient p (hPa) | Minimum altitude (km) | Maximum altitude (km) | Final bag pressure absolute (hPa) | Estimated sampled volume (mL STP) |
|---|---|---|---|---|---|---|
| KRN – BL1 | 10.7 | 18.8 | 27.0 | 30.7 | 48.8 | 200 |
| KRN – BL2 | 19.3 | 36.0 | 22.8 | 26.8 | 101.0 | 400 |
| KRN – BL8 | 27.1 | 48.8 | 20.8 | 24.6 | 99.3 | 400 |
| KRN – BL3 | 37.2 | 54.4 | 20.1 | 22.5 | 186.9 | 750 |
| KRN – BL9 | 56.2 | 75.2 | 18.0 | 19.0 | 166.6 | 670 |
| KRN – BL4 | 65.1 | 78.9 | 17.7 | 19.0 | 275.7 | 1100 |
| KRN – BL5 | 97.8 | 105.9 | 15.9 | 16.4 | 303.5 | 1210 |
| KRN – BL6 | 126.1 | 131.4 | 14.5 | 14.7 | 252.2 | 1010 |
| SOD2 – L1 | 132.3 | 143.6 | 14.1 | 14.7 | 258.0 | 650 |
| SOD2 – L2 | 80.0 | 95.2 | 16.8 | 18.0 | 236.4 | 590 |
| SOD2 – L3 | 39.2 | 46.5 | 21.5 | 22.7 | 89.9 | 230 |
| SOD3 – L1 | 133.7 | 141.8 | 14.2 | 14.6 | 262.5 | 660 |
| SOD3 – L2 | 82.1 | 93.8 | 17.0 | 17.9 | 257.3 | 640 |



| | | | | | |
|---|---|---|---|---|---|
| SOD3 – L3 | 35.0 | 46.5 | 21.6 | 23.5 | 170.4 | 430 |
| SOD4 – L1 | 130.2 | 141.0 | 14.3 | 14.6 | 261.0 | 650 |
| SOD4 – L2 | 79.6 | 94.3 | 17.0 | 18.0 | 242.2 | 610 |
| SOD4 – L4 | 18.6 | 28.7 | 24.8 | 27.7 | 76.5 | 190 |

## 2.2  Quantum Cascade Laser Spectrometer (QCLS)

The trace gas analyser used to perform COS measurements in all campaigns is a dual laser QCLS by Aerodyne Research Inc.
(Billerica, MA, USA), operating in the mid-infrared frequencies. This technique was firstly introduced for COS measurements
by Stimler et al. (2009) and further developed by Kooijmans et al. (2016). The QCLS employed in this study has also been
used in previous UAV- and aircraft-borne tropospheric active AirCore measurements for $CH_4$ and $N_2O$ (Tong et al., 2023;
Vinković et al., 2022), as well as in situ tropospheric COS observations (Zanchetta et al., 2023). The QCLS can measure $CH_4$,
$CO_2$, $N_2O$, CO, COS, $H_2O$, and $O_3$ simultaneously. Its cavity is controlled at a temperature of 298 K and a pressure of ~66 hPa
(50 Torr). The QCLS measures at a constant mass flow of 50 mL min$^{-1}$ and the measured data is output at 1 Hz. For COS, the
precision (1σ) falls usually between 15 and 25 ppt at 1 Hz, depending on the laboratory conditions (e.g., ambient temperature
stability). The cell of the QCLS has a volume of 150 mL, which at 50 Torr corresponds to an effective cavity volume of ~10
mL. The spatial resolution of AirCore measurements in this configuration is roughly 2000 m at 28 km, 300 m at 15 km and
200 m at 10 km altitude, which resembles the resolution ranges presented by Membrive et al. (2017). The QCLS is controlled
with a custom-made frontend, operated via a dedicated software, which allows switching between different inlets without
causing changes in pressure through the system. Overall, the instrument achieves a precision better than 0.6 ppb for $CH_4$, 0.2
ppm for $CO_2$, 0.12 ppb for $N_2O$, 1 ppb for CO, 20 ppt for COS, 20 ppm for $H_2O$, and 100 ppb for $O_3$.

## 2.3  Datasets for COS profiles comparison and validation

For validation purposes, our COS profiles were compared to the SPIRALE results reported by Krysztofiak et al. (2015), as
well as a selection from the Atmospheric Chemistry Experiment – Fourier Transform Spectrometer (ACE-FTS) dataset
(Bernath, 2005; Velazco et al., 2011).

### 2.4.1 SPIRALE

The in situ balloon-borne SPIRALE spectrometer presented by Krysztofiak et al. (2015) was deployed in two flights at polar
latitudes (Kiruna, Sweden, 67°53' N 21°04' E) in 2009 and 2011. The resulting COS profiles cover altitudes between 14.3 –
21.6 km and 18.5 – 22.0 km, respectively. Given the different altitudinal resolution, both the measured AirCore profiles and
the SPIRALE results were averaged in 0.5 km bins to calculate the difference between COS mole fractions. A comparison
between AirCore and SPIRALE COS profiles is shown in Figure 3.





### 2.4.2 ACE-FTS

ACE-FTS is a satellite-borne spectrometer measuring altitude profile information for temperature, pressure, and mole fractions

of several gas species (Bernath, 2005; Glatthor et al., 2017; Velazco et al., 2011), including COS, by sun occultation. Each profile of ACE-FTS contains 1 km resolution data from 0-150 km altitude. To have a comparable dataset with the measured AirCore profiles, COS ACE-FTS observations realised between June and September in the 2012 – 2024 period were selected, with latitudes ranging between 45-49° N for TRN and 65-69° N for KRN and SOD, resulting in 502 and 1681 COS profiles, respectively. These selected profiles were then averaged for both latitudinal ranges. The COS profiles measured with AirCore

in this study are averaged over 1 km intervals to obtain a comparable dataset. The resulting averaged profiles and their comparison with the observed AirCore profiles are shown in Figure 5.

### 3.  Results

### 3.1  COS observations

Figure 1 shows the COS profiles measured from AirCores and the LISA/BigLISA samples of the campaigns reported in Table

1. For all campaigns, different inlets were deployed. Different inlet configurations are reported in Table 1 and in the figure captions.

The tropospheric COS mole fractions vary from flight to flight, ranging between about 400 to 500 ppt. The thermal tropopause height (World Meteorological Organization, 1957) was between about 10.6 and 11.5 km at mid latitudes (TRN, June 2019),

and between 9.9 and 10.9 km at polar latitudes (KRN, August 2021 and SOD, 2023), respectively (see Fig. S7-S15 in Sect. S3 of the supplement). The COS stratospheric sink is clearly noticeable for all campaigns above the tropopause and is further discussed in Sect. 4.1. At polar latitudes, the COS mole fraction decreases from an all-campaigns average of $421 \pm 26$ ppt below 17 km to $180 \pm 15$ ppt between 20 and 22 km. At mid-latitudes, the mole fractions decrease from a campaign average of $488 \pm 12$ ppt below 17 km to $220 \pm 59$ ppt between 20 and 22 km.


As reported in Table 3, the BigLISA samples measured in KRN in 2021 show consistently higher mole fractions when compared to the AirCore profiles. By contrast, the LISA samples measured in SOD in general show good agreement with the continuous profiles. The largest difference was found in SOD3, between a collection of free-inlet LISA samples and an AirCore equipped with a cotton scrubber obtained from the same flight. The possible explanations for these differences are discussed

in Sect. 4.1.2.





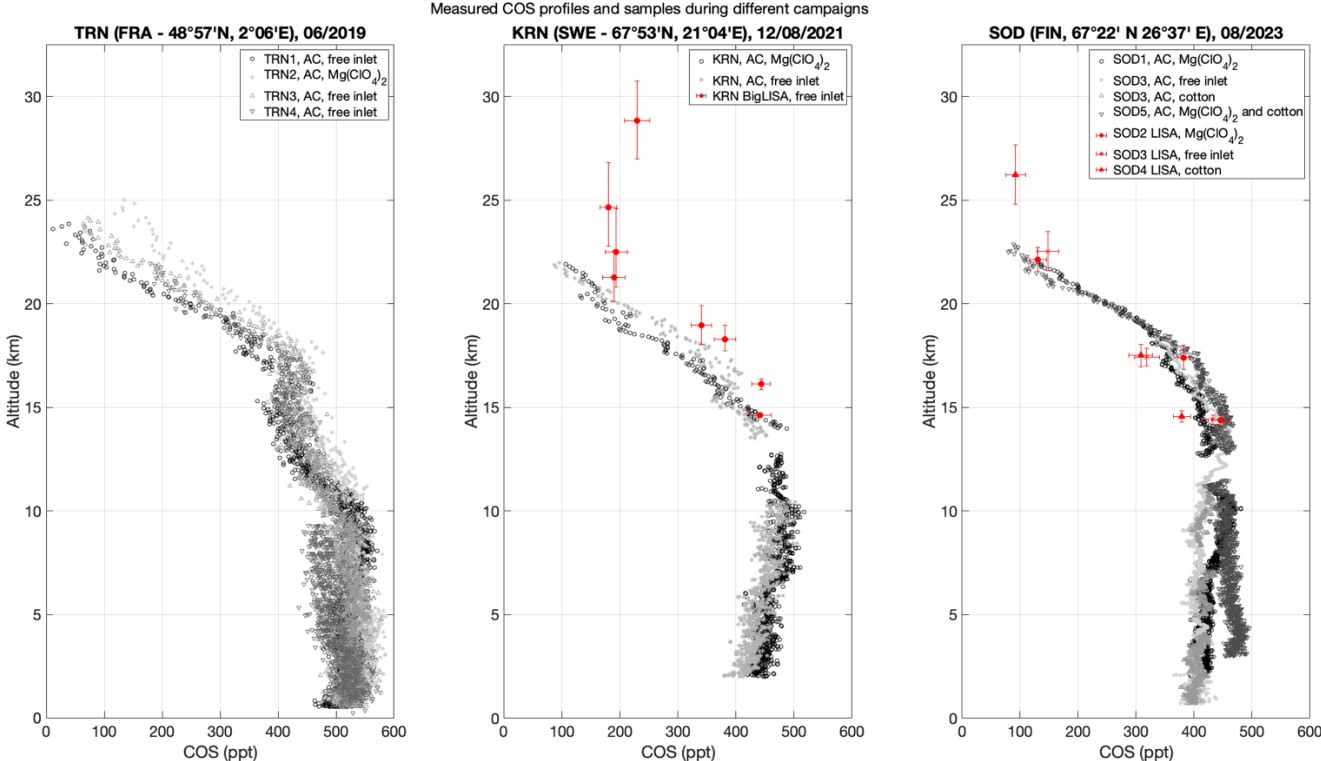

**Figure 1: measured COS AirCore (AC) profiles and LISA/BigLISA samples in different sampling campaigns.**

**Table 3: COS mole fraction difference between AirCore and (Big)LISA samples that flew on the same days. The AirCore COS mole fraction is calculated as the COS average over (Big)LISA's sampling altitude range.**

| (Big)LISA sample name | Altitude range (km) | ΔCOS (ppt) AirCore – (Big)LISA | Inlets features | Average ΔCOS (ppt) |
|---|---|---|---|---|
| KRN – BL3 | 20.1 – 22.5 | -48 | AirCore: $Mg(ClO_4)_2$ dryer BigLISA: $Mg(ClO_4)_2$ dryer | -80 ± 50 |
| KRN – BL9 | 18.0 – 19.9 | -125 | | |
| KRN – BL4 | 17.7 – 19.0 | -133 | | |
| KRN – BL5 | 15.9 – 16.4 | -78 | | |
| KRN – BL6 | 14.5 – 14.7 | -16 | | |
| KRN – BL3 | 20.1 – 22.5 | -52 | AirCore: free inlet BigLISA: $Mg(ClO_4)_2$ dryer | -53 ± 20 |
| KRN – BL9 | 18.0 – 19.9 | -68 | | |
| KRN – BL4 | 17.7 – 19.0 | -70 | | |
| KRN – BL5 | 15.9 – 16.4 | -53 | | |
| KRN – BL6 | 14.5 – 14.7 | -21 | | |





| | | | | |
|---|---|---|---|---|
| SOD1 – L1 | 14.1 – 14.6 | -32 | AirCore: Mg(ClO₄)₂ dryer<br><br>LISA: Mg(ClO₄)₂ dryer | -16 ± 24 |
| SOD1 – L2 | 16.8 – 18.0 | -27 | | |
| SOD1 - L3 | 21.5 – 22.7 | 11 | | |
| SOD3 – L1 | 14.2 – 14.6 | 17 | AirCore: free inlet<br><br>LISA: free inlet | 31 ± 20 |
| SOD3 – L2 | 17.0 – 17.9 | 46 | | |
| SOD3 – L3 | 21.6 – 23.5 | No AirCore data | | |
| SOD3 – L1 | 14.2 – 14.6 | 25 | AirCore: cotton, without dryer<br><br>LISA: free inlet | / |
| SOD3 – L2 | 17.0 – 17.9 | No AirCore data | | |
| SOD3 – L3 | 21.6 – 23.5 | No AirCore data | | |

## 4. Discussion

### 4.1 Collected COS profiles and (Big)LISA samples

#### 4.1.1 COS profiles

The AirCore data presented in Figure 1 show continuously sampled stratospheric COS profiles collected with balloon-borne instruments. Previous stratospheric or Upper Troposphere/Lowermost Stratosphere (UT/LMS) observations were realised with discrete whole-air sampling (Engel and Schmidt, 1994; Karu et al., 2023), in situ spectrometers (Gurganus et al., 2024 (preprint); Kloss et al., 2021; Krysztofiak et al., 2015; Leung et al., 2002; Toon et al., 2018; Wofsy et al., 2017; Wofsy, 2011) or remote sensing (Glatthor et al., 2017; Velazco et al., 2011; Yousefi et al., 2019). LISA and BigLISA represent lightweight

additions to these methods. For validation purposes, AirCore, LISA and BigLISA observations have been compared with SPIRALE in situ observation (Krysztofiak et al., 2015) and the ACE-FTS remote sensing observations (Bernath, 2005; Velazco et al., 2011).

The occasional data gaps shown in the AirCore profiles have different reasons, as summarised in Table 4. TRN4 and the KRN

AirCore equipped with Mg(ClO₄)₂ had tubing of two different diameters, which were connected to each other using a sleeve adapter into which they were glued using Loctite Super Attak glue. SOD1 and SOD5 AirCores, instead, were equipped with differential pressure sensors and showed signs of COS contamination at the sensors' location. SOD3 AirCore showed COS spikes related to the O₂ altitude-mapping technique employed in one of its halves. COS outgassing from sulfur-containing polymers, in particular rubbers, has been reported in several studies (Cadle and Williams, 1978; Levine et al., 2023; Pos and

Berresheim, 1993). We speculate that the polymers constituting the glue, or components of valves (e.g., O-rings) and differential pressure sensors, may have caused the COS outgassing. In the case of SOD3, another possibility could be a direct reaction of other gas species with O₂.



Nonetheless, the AirCore profiles we presented are similar to observations reported by previous studies: in TRN, COS mole
fraction first decreases from tropospheric values of about 510 ppt up to around 10.5 km, to about 420 ppt at 17 km. Then, it
undergoes a faster decrease to 92 – 203 ppt at 22 km. This is consistent with observations reported in previous studies at
comparable latitudes (Leung et al., 2002; Toon et al., 2018).

At polar latitudes, we observed larger tropospheric variability. The profiles showed tropospheric COS mole fractions ranges
between 400 – 470 ppt. Glatthor et al. (2017) reported values at 5 km altitude as low as 330 ppt for COS, during northern
summer months at polar latitudes between 2003 – 2012, while Toon et al. (2018) showed a value around 410 ppt under similar
conditions. Regarding the stratospheric part of the profile, we observed slight increases or stable values up to 13 – 15 km range
(350 – 450 ppt), followed by decreases down to 95 – 130 ppt around 22 km. Most flights from the SOD campaign, in particular
SOD1 and SOD5, were characterised by variable lapse rates above the tropopause (see Sect. S3 in the Supplement) which we
suspect may be an indicator for COS convective transport and mixing in the lowermost stratosphere. Nevertheless, the observed
stratospheric COS range seems consistent with previous studies. Leung et al. (2002) reported 440 ppt below 14 km, decreasing
to 120 ppt at 22 km. Glatthor et al. (2017) reports around 490 ppt at 13 km, followed by a decrease to 140 ppt at 22 km.
Krysztofiak et al. (2015) reported 420 ± 100 ppt COS below 17 km, decreasing to 150 ppt at 22 km. A more quantitative
comparison between our profiles and Krysztofiak et al. (2015) and ACE-FTS (Bernath, 2005; Velazco et al., 2011; Yousefi et
al., 2019) observations is presented in Sect. 4.2 and 4.3, respectively.

### 4.1.2 (Big)LISA samples

The samples from KRN BigLISA show clear signs of contamination, in particular above 20 km (Figure 1). We believe this
may be due to COS outgassing from some plastic components of the MLF bags used to collect the samples, such as the O-ring
in the valves, or simply tropospheric air that remained trapped inside the bag. Although the deployed bags are indicated as
suitable for sulfur compounds, they are not recommended for low-ppm volatile organic compounds due to background levels
(Sigma Aldrich, 2025). This might have also influenced our COS measurements, perhaps due to spectroscopic effects. The
contamination appears to be inversely proportional to the sampling pressure and the collected sample volume. Unfortunately,
it has not been possible to assess the cause of this contamination precisely. Given these circumstances and the impossibility of
applying any correction to these results, BigLISA will be left out of the discussion and comparisons with other datasets.


However, as described in Sect. 2.1.2, during the SOD campaign we introduced a pre-treatment technique that has solved this
issue for COS, based on previous laboratory tests. Filling and vacuuming the bags with a stratospheric-mimicking gas seemed
to have reduced the contamination significantly for most LISA samples, as previously reported in Figure 1 and Table 3. When
LISA flew on the same balloon as one of our AirCores (e.g. SOD1, SOD3), the largest average difference of 31 ± 15 ppt was
found when both instruments were flown with a free inlet. Although some variability can be observed between different LISA





samples at similar altitudes, their COS mole fraction falls well within the range of the AirCore profiles (see Sect. 4.1.3, 4.2 and 4.3).

### 4.1.3 Datasets consistency

Figure 2 shows the measured COS AirCore profiles and LISA samples from all campaigns, plotted against altitude above
tropopause (see Sect. S3 in the Supplement). TRN1 is not presented in this figure, since the tropopause height could not be estimated due to missing temperature and relative humidity. Differences of up to ~100 ppt can be clearly seen between the measured profiles. However, these differences are not constant with altitude and do not show any clear trend over the time span of the campaigns. Moreover, these differences do not show any clear relationship with the different inlets employed.

The most likely explanation for differences between the measured profiles may reside in stratospheric horizontal transport
from different latitudes (Toon et al., 2018). The tight correlation between $CH_4$ and $N_2O$ (Sect. S4 in the Supplement) suggests that the day-to-day variability can likely be ascribed to atmospheric transport (Kondo et al., 1996; Plumb, 2007; Plumb and Ko, 1992). Moreover, long-term changes in COS seasonal cycle, sources and sinks (Belviso et al., 2022; Sturges et al., 2001) may affect its stratospheric abundance.

Other possible causes of the differences between the profiles may reside in a combination of instrumental uncertainties, altitude
mapping algorithms, sample loss after landing, contaminations and instrumental features (e.g., inlets, different air mixing in the AirCore coil during sampling). Overall, it is difficult to assess the contribution of each of these parameters quantitatively. Instrumental uncertainties and altitude mapping are self-consistent for our results. No quantification of sample loss is available, but given the AirCore design, this should mostly affect the tropospheric part of the profiles. Some clear contaminations affecting AirCore profiles (Table 4) and (Big)LISA samples were marked as outliers, but it is still possible that more subtle
effects affected the samples. These may include mixing with dead volumes of tropospheric air or fill gas, impurities in the deployed scrubbers or effects due to the instrumental components (e.g., O-rings, tubing). However, these differences shall remain marginal compared to the aforementioned day-to-day variability and long-term trends.

**Table 4: data gaps and their causes in different AirCore profiles.**

| Flight identifier and AirCore inlet | Data gaps | Contamination points along the coil |
|---|---|---|
| TRN4 – $Mg(ClO_4)_2$ | 9.2 – 13.9 km | Glue connection between tubes of different diameter in the coil |
| KRN – $Mg(ClO_4)_2$ | 12.7 – 13.9 km | Glue connection between tubes of different diameter in the coil |
| SOD1 – $Mg(ClO_4)_2$ | 11.9 – 12.8 km | Differential pressure sensors |





| | 4.0 – 5.0 km | |
|---|---|---|
| SOD3 – cotton | 7.4 – 8.3 km | Valve employed in $O_2$ spiking technique to retrieve altitude, or reactions of other species with $O_2$ itself |
| | 11.1 – 12.3 km | |
| | 16.5 – 17.9 km | |
| | 20.4 – 21.3 km | |
| SOD5 – $Mg(ClO_4)_2$ and cotton | 12.4 – 13.7 km | Differential pressure sensors |


The two SOD3 profiles sampled with University of Bern's double-sided AirCore are of particular interest, since one side was equipped with a cotton-based $O_3$ scrubber and the other side was left with a free inlet. As reported in Table 4, the spiking technique deployed by the University of Bern caused COS anomalies in one of the profiles. Moreover, as explained in Sect. 2.1.1, the effect of fill gas was amplified for this sampler: the correlation between $CH_4$ and $N_2O$ (Fig. S18 in the Supplement)

shows a clear deviation for $N_2O$ mole fractions lower than 280 ppb (corresponding to an altitude of roughly 18 km). This prevented a reasonable analysis of possible effects of $O_3$ and/or of the efficiency of the cotton scrubber. However, the resulting (shortened) COS profiles show very good accordance with the profiles obtained from other AirCores and small differences ranging between 17 – 46 ppt with the comparable LISA samples (Table 3).

Since it has been shown that $O_3$ daily variability can affect oxidizable species (Dirksen et al., 2011; Li et al., 2021), we also speculated that part of the differences between the measured profiles may be ascribed to a daily cycle, similarly to what happens for $O_3$ (Frith et al., 2020; Li et al., 2021; Schranz et al., 2018; Studer et al., 2014). Suggestions of a correlation between COS and $O_3$ stratospheric chemistry and abundance have been presented in previous studies (Engel and Schmidt, 1994). OH• and O•, known to cause COS loss (Brühl et al., 2012; Chin and Davis, 1995; Krysztofiak et al., 2015), are part of the Chapman

cycle of $O_3$ (Frith et al., 2020; Studer et al., 2014). $O_3$ daily variability is reported to change depending on seasonality (daily patterns in solar radiation, photolysis), latitude, temperature and atmospheric pressure level (Frith et al., 2020; Studer et al., 2014). However, reaction kinetics and the current estimates of stratospheric COS sinks of $50 \pm 15$ GgS yr$^{-1}$ due to photolysis (Whelan et al., 2018) do not support the hypothesis of a detectable daily cycle with the methods presented in this study. On top of that, since we are uncertain about the effectiveness of the cotton-based scrubbers deployed during our campaigns and

we have no $O_3$ measurements, we cannot support this hypothesis.

Overall, we consider our results to be a trustworthy representation of the COS stratospheric conditions during sampling. In fact, in spite of the variability, all profiles show similar trends when compared with one another and with the LISA samples, too. All measurements show the expected stratospheric sink clearly. Their agreement with SPIRALE and ACE-FTS

observations will be discussed in Sect. 4.2 and 4.3, respectively.





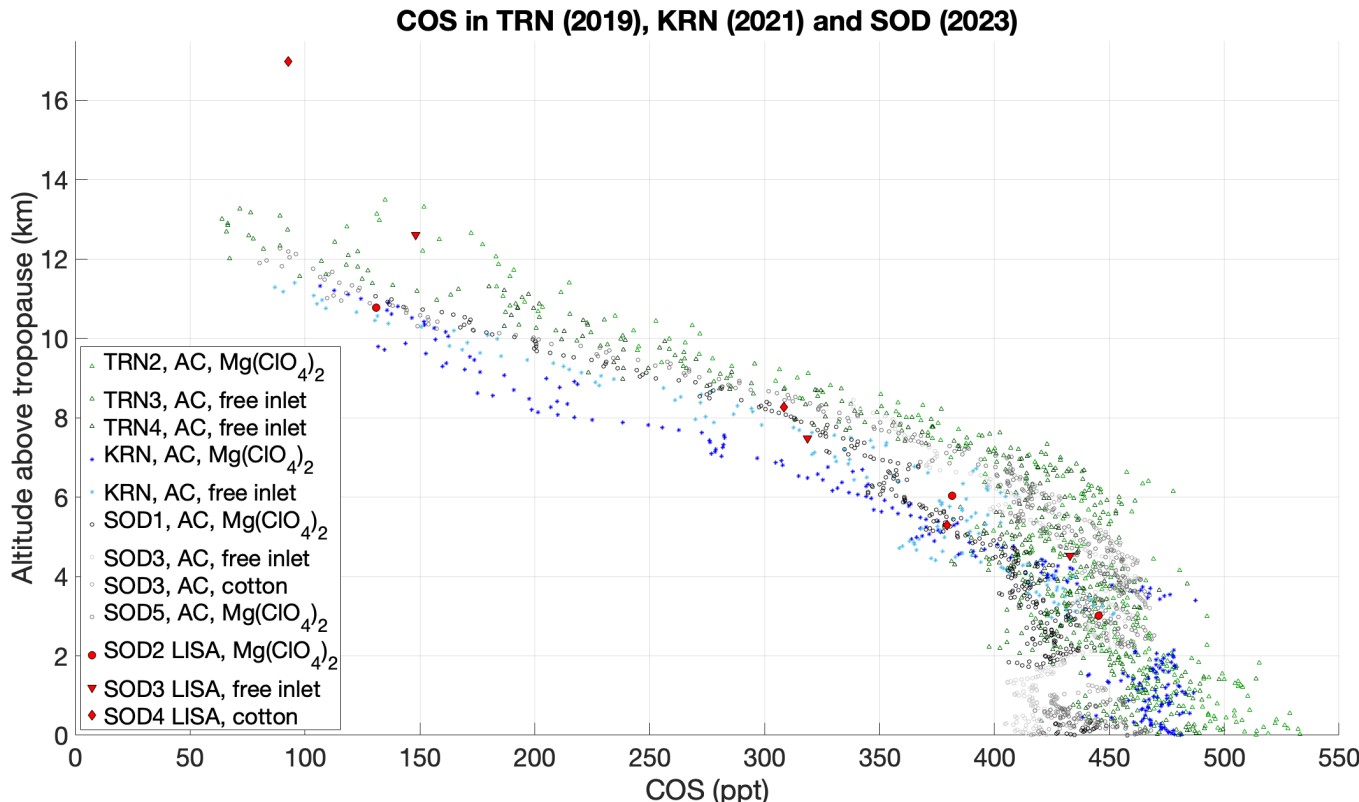

Figure 2: COS profiles and samples from all campaigns, plotted against altitude above tropopause.

**4.2 Comparison with SPIRALE observations**

Figure 3 shows a comparison plot of AirCore and LISA observations with the in situ SPIRALE COS observations realised by
Krysztofiak et al. (2015) in 2009 and 2011. It is evident that SPIRALE observations are only available between 14.5 and 22
km. These observations have a higher spatial resolution over the vertical column (3 to 5 m), but are associated to lower
precision compared to AirCore COS measurements analysed by QCLS. Moreover, both techniques measure profiles that are
specific for the location and the time where the measurement occurs. In this case, both studies have data collected at polar

latitudes, although at different times of the year and, most importantly, between 10 and 14 years apart from each other. To
realise a meaningful comparison, both AirCore profiles and SPIRALE observations have been averaged over 0.5 km bins.
Using the averaged bins, a linear regression was performed and the difference between AirCore and SPIRALE was calculated.
As shown in Figure 3, most AirCore profiles and all the LISA samples collected at altitudes comparable to the SPIRALE
datasets fall within the uncertainty range of both SPIRALE profiles. The horizontal error bars reported in Figure 3a represent

the uncertainty of QCLS measurements, while the ones in Fig. 3b and 3c are the standard deviation obtained from the averaging
of SPIRALE over LISA's sampling altitude intervals and AirCore profiles over 0.5 km altitude intervals. The shaded area in
the second and third panel correspond to SPIRALE's errors.



Figure 6 and Table 5 show plots and results of the linear regressions of AirCore and LISA vs. SPIRALE observations in 2009
and 2011, respectively. The linear regression of averaged KRN and SOD observations against SPIRALE's 2009 observations
leads to $R^2$ values above 0.7 for both AirCores and LISA, with a slope of and $0.921 \pm 0.061$ and $0.892 \pm 0.230$ respectively.
This, in spite of the time span between the two experiments, indicates a high correlation between the methods, with AirCore
results being approximately 8% lower than SPIRALE's 2009 observations, while LISA's regression slope suggests a 11% bias
but does not significantly different from 1. Between 15.5 and 17.5 km, differences between 40 – 110 ppt can be seen between
both KRN and SOD AirCore profiles and the 2009 SPIRALE. However, in this range (in particular between 15.8 and 16.2
km), SPIRALE measured a COS spike that reached up to 577 ppt, a rather unusual mole fraction for these altitudes, which is
reflected also in a previous comparison between SPIRALE and ACE-FTS (see Fig. 5 in Krysztofiak et al., 2015). Moreover,
as reported in Fig. 5 of Krysztofiak et al. (2015), SPIRALE results fall generally above the averaged ACE-FTS observations.
Unfortunately, only COS measurements are available from SPIRALE and it is not possible to verify this idea with observations
of other tracers. Nevertheless, considering day-to-day variability (e.g. discrepancies in tropopause height, or the above-cited
air transport), the 12 to 14 years differences between the campaigns and the long-term trends occurred in this time span
(Bernath et al., 2020; Glatthor et al., 2017; Hannigan et al., 2022; Kremser et al., 2015; Lejeune et al., 2017), the linear
regressions suggest a strong agreement between the datasets, with the AirCore and LISA results being generally lower than
SPIRALE results in 2009.


With regard to SPIRALE observations in 2011, the linear regression could be only performed on 27 averaged AirCore COS
mole fractions from the KRN and SOD campaigns and resulted in a weaker correlation between this dataset and our results
compared to the 2009 observations. This could be due to the limited altitudinal range of these observations and due to the
different sampling season (April for this SPIRALE measurements, August for our campaigns), on top of the reasons mentioned
earlier in the text for the differences between our data and the SPIRALE 2009 observations. Nevertheless, at polar latitudes
we observe COS mole fraction decreasing from an all-campaign average of $421 \pm 26$ ppt below 17 km to $170 \pm 41$ ppt between
$20 - 22$ km. This is comparable to the decrease from 460 to 150 ppt reported by Krysztofiak et al. (2015) and from 440 to 120
ppt reported by and Leung et al. (2002).





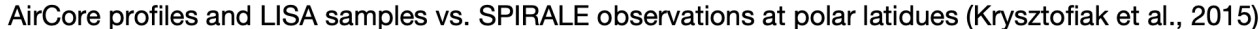

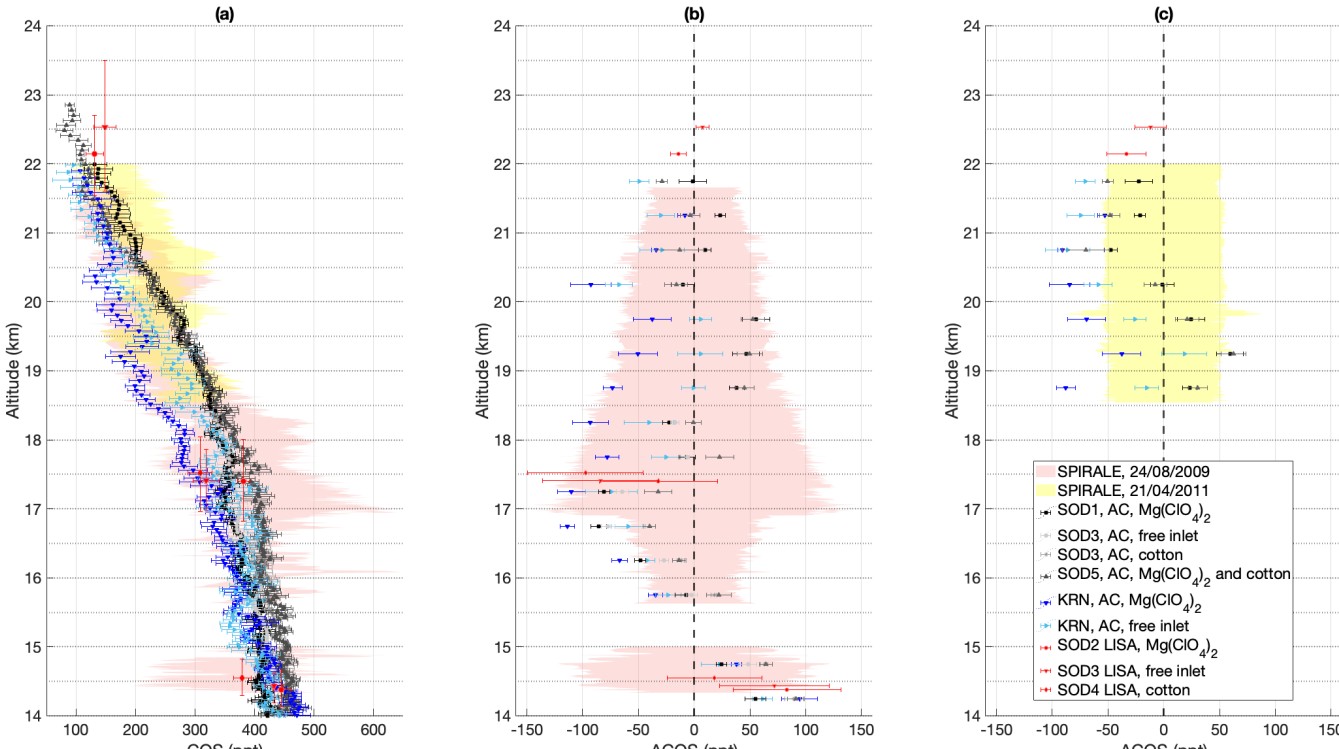

**Figure 3: measured COS profiles and LISA samples at polar latitudes (Kiruna - KRN, Sodankylä - SOD), compared with the SPIRALE's in situ observations from flights in KRN (red and yellow shading representing COS ± std. dev.), presented by Krysztofiak et al. (2015). The dotted horizontal lines signal the intervals within which the average COS mole fraction is calculated. Panel (a) shows the measured profiles, while panels (b) and (c) show the difference between the measured AirCore profiles and the SPIRALE results. The horizontal error bars reported in panel (a) represent the uncertainty of QCLS measurements. The error bars in panels (b) and (c) are the standard deviation obtained from the averaging of SPIRALE over LISA's sampling altitude intervals, and over AirCore profiles over 0.5 km altitude intervals**

### 4.3 Comparison with ACE-FTS observations

Figure 4 and Figure 5 show the comparison of the measured TRN, KRN and SOD AirCore profiles with the ACE-FTS averaged observations, while Figure 7 and Table 5 show the results of the linear regression models of both campaigns. The agreement between the averaged ACE-FTS profiles and the AirCore profiles is rather coherent at both mid and polar latitudes. The TRN data shows good agreement between ACE-FTS and AirCore profiles, in particular in the stratospheric part of the profile, where most differences fall within the ± 50 ppt range. The biggest discrepancies are found in the tropospheric part, where seasonal variabilities and daily variations are more pronounced. At polar latitudes, the profiles measured in KRN in 2021 show better agreement with the ACE-FTS results, while the SOD measurements generally resulted in higher COS mole fractions above 15 km altitude, with differences up to ~80 ppt. This may be due to specific conditions during the period when the flights were performed, such as different atmospheric transport patterns, or the uncertainties in the AirCore methodology described in Sect. 4.1.3. Similarly to what has been described in Sect. 4.2, linear regressions between ACE-FTS and AirCore and LISA samples





were performed to quantify the level of agreement between the results. In this case, AirCore profiles were averaged over 1 km intervals to make them comparable to the ACE-FTS resolution.


The linear regressions (Figure 7) between polar summer ACE-FTS average and AirCore profiles from all the campaigns at polar latitudes resulted in high correlations ($R^2$ values > 0.9) and a slope signalling a difference of roughly 5% between the two methods. Consistently, the linear model applied to the mid-latitude ACE-FTS selection and the TRN AirCore profiles resulted in high correlation and a difference of roughly 5%. The regression between SOD LISA results and the ACE-FTS

instead suggests a difference of roughly 10% with a slightly lower correlation which may be due a higher COS measured in the highest measured sample not flagged as outlier, SOD4 – L4, compared to the ACE-FTS average (93 ppt against 29 ppt). The removal of this sample from the regression leads to a difference of less than 7%. The resulting intercept of the regression between polar ACE-FTS data and polar LISA samples is significantly higher than 0, suggesting slightly higher estimates of COS mole fraction with LISA compared to ACE-FTS when approaching low COS levels. However, the estimated slopes

indicate that the data difference with ACE-FTS over altitude is less than 5% for AirCores and roughly 10% for LISA.

Most profiles show higher COS mole fractions in the 14 – 24 km range at polar latitudes and in the 16 – 24 km range at mid-latitudes when compared to ACE-FTS averages. Glatthor et al. (2017) reported a difference up to 100 ppt between COS mole fraction retrieved from MIPAS remote sensing and the ACE-FTS ones between 13 – 16 km. Velazco et al. (2011) found COS

mole fractions 15% higher than in situ spectrometry and ACE-FTS profiles, while Krysztofiak et al. (2015) reported consistency within 11% at polar latitudes and a positive difference of 15 – 20% at mid-latitudes, taking into account both instrumental uncertainties. The discrepancy we observe between AirCore profiles and ACE-FTS is in the range of 80 ppt and maximises between 17 – 22 km. Nonetheless, some profiles show lower COS mole fractions in the same ranges. Therefore, the differences are either due to day-to-day variability, or due to instrumental issues, such as different scrubbers that were

employed in different flights.





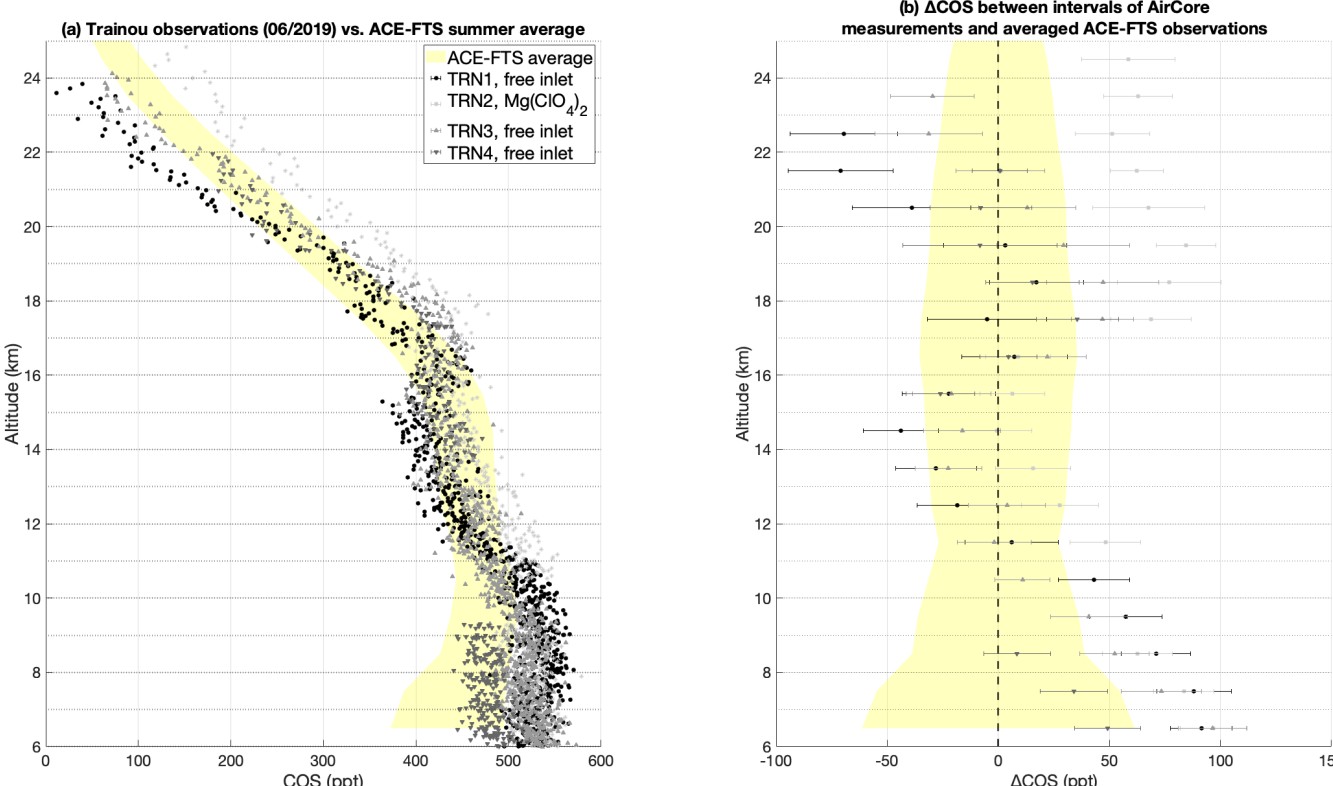

**Figure 4: comparison of the AirCore profiles in Trainou (TRN) with ACE-FTS average over summer months between 45 – 49° N. In panel (a), the shaded yellow area represents the averaged ACE-FTS results ± 1σ. The dotted horizontal lines signal the intervals within which the average is calculated. In panel (b), the shaded area corresponds to ACE-FTS 0 difference, ± 1σ.**




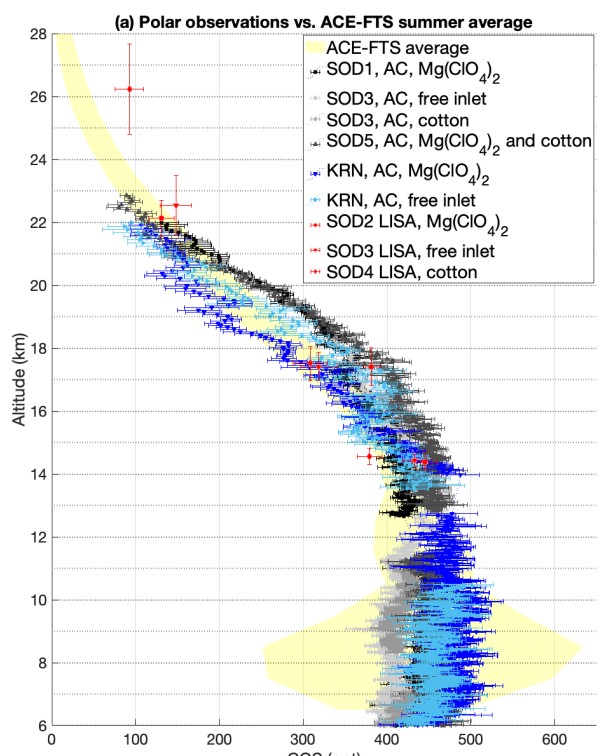
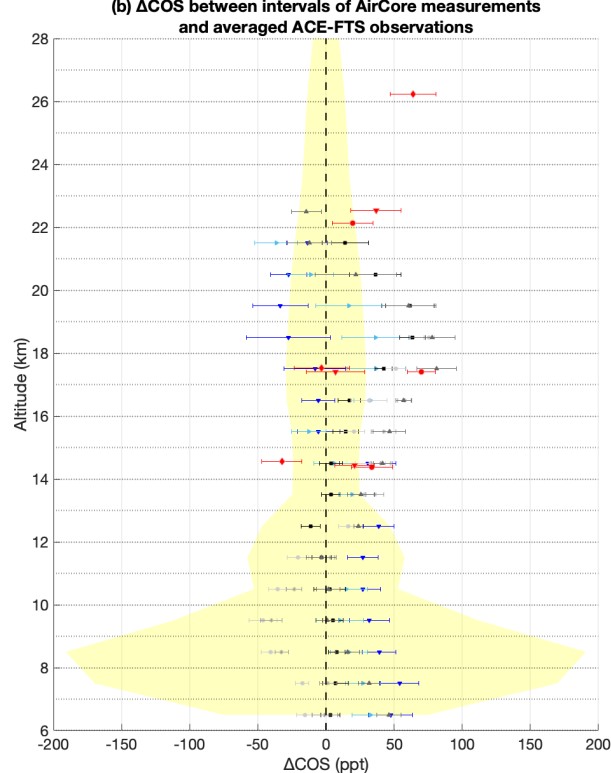

**Figure 5: comparison of the AirCore profiles (Kiruna - KRN, Sodankylä - SOD) and LISA samples with ACE-FTS average over summer months at 65 – 69° N. In panel (a), the shaded yellow area represents the averaged ACE-FTS results ± 1σ. The dotted horizontal lines signal the intervals within which the AirCores' average is calculated. In panel (b), the shaded area corresponds to ACE-FTS 0 difference, ± 1σ.**




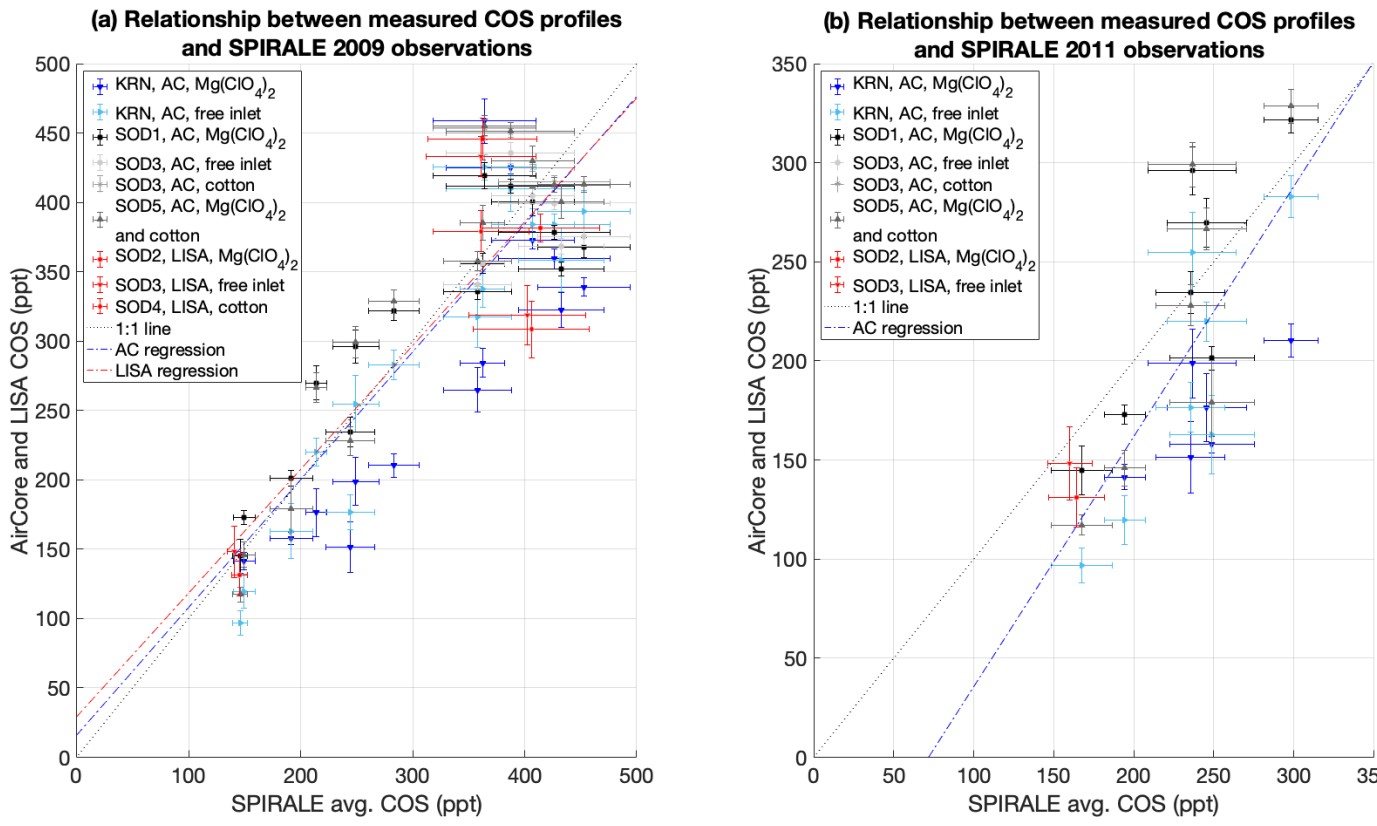

Figure 6: linear regressions between SPIRALE observations (panel a, 2009 and panel b, 2011) and AirCore and LISA samples. The dotted black line represents the 1:1 line. In panel (b) the regression between LISA samples and SPIRALE averaged COS mole fractions was not possible, since only two data points collected at similar altitudes were comparable. The statistical values related to
these regressions are reported in Table 5.





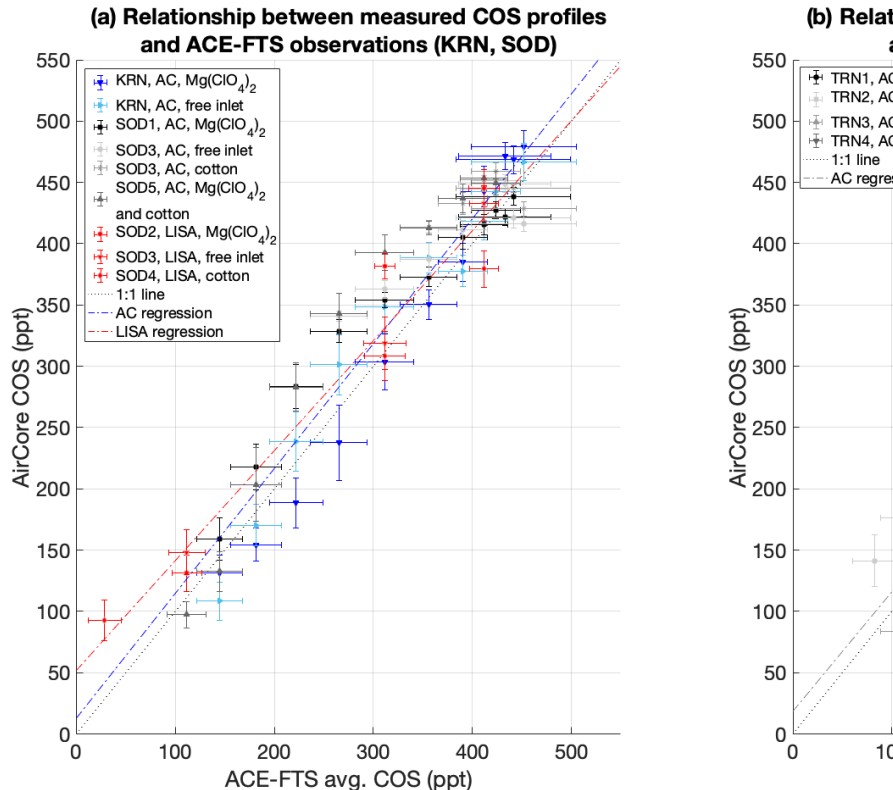

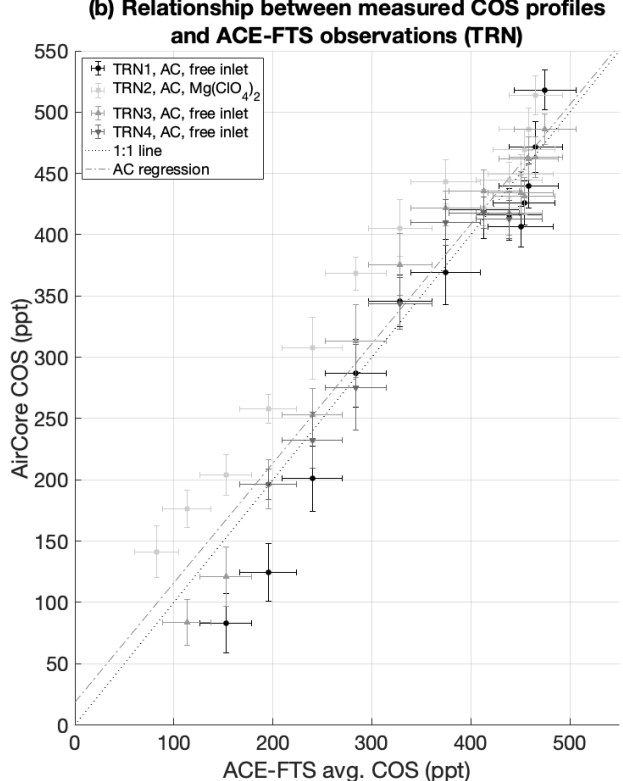

**Figure 7: linear regressions between ACE-FTS averaged observations andAirCore and LISA samples at polar latitudes (KRN and SOD, panel a) and at mid latitudes (TRN, panel b). The dotted black line represents the 1:1 line. The statistical values related to these regressions are reported in Table 5.**

**Table 5: summary of the performed linear regressions between measured samples and previous COS remote sensing and in situ spectrometry observations (see Figure 6 and Figure 7).**

| Independent variable | Dependent variable | Slope | Intercept | Number of observations | $R^2$ |
|---|---|---|---|---|---|
| SPIRALE 2009 | KRN and SOD AirCore observations | $0.921 \pm 0.061$ | $16 \pm 21$ | 71 | 0.766 |
| SPIRALE 2009 | SOD LISA observations | $0.892 \pm 0.230$ | $29 \pm 79$ | 8 | 0.714 |
| SPIRALE 2011 | KRN and SOD AirCore observations | $1.26 \pm 0.23$ | $-90 \pm 56$ | 27 | 0.533 |



| | | | | | |
|---|---|---|---|---|---|
| SPIRALE 2011 | SOD LISA observations | - | - | 2 | - |
| ACE-FTS "TRN" average | TRN AirCore observations | $0.975 \pm 0.045$ | $18 \pm 16$ | 48 | 0.911 |
| ACE-FTS "polar" average | KRN and SOD AirCore observations | $1.019 \pm 0.040$ | $13 \pm 14$ | 57 | 0.923 |
| ACE-FTS "polar" average | SOD LISA observations | $0.897 \pm 0.072$ | $52 \pm 22$ | 9 | 0.957 |

## 5. Conclusion

This study presented in situ stratospheric COS observations based on collected air samples using two new techniques, AirCore and (Big)LISA samplers. The collected continuous and discrete stratospheric samples were analysed with a QCLS in the
laboratory. The results obtained with both techniques closely resemble the stratospheric trends retrieved from previous discrete samples and in situ spectroscopic observations. Moreover, we found less than 5% difference between AirCore data and averaged ACE-FTS data obtained with remote sensing, although we observe higher COS estimations when approaching low COS abundances when compared to ACE-FTS. We found that, when deploying MLF bags to measure COS, it is necessary to pre-treat the bags before flight to prevent COS contamination due to outgassing from the polymers constituting the bag. We
also found that cotton-based $O_3$ scrubbers may have limited efficiency, especially when cotton has been in contact with the air for several months. Squalene-based scrubbers showed excellent $O_3$ scrubbing performances and seemed to have no effect on COS abundance and may become a valuable addition to stratospheric samplers that require $O_3$ removal.

We found no clear evidence that stratospheric $O_3$ causes positive or negative biases in COS measurements, since no repeatable
differences were found in our samples while deploying different sorts of inlets. However, the observed COS mole fractions showed some day-to-day variability which may be ascribed to stratospheric transport or instrumental biases. Consistent differences in COS profiles point to observed transport variability, hypothesis that may be corroborated by modelling efforts. The investigation on the effects of $O_3$ on air samples, in particular containing reduced sulfur species, could be facilitated by the deployment of the squalene-based scrubbers.




**Appendix A**

**Ozone scrubbers**

$O_3$ is a reactive gas species that can be found at mole fractions up to about 8 ppm in the stratosphere, where it is formed by the interaction of atmospheric $O_2$ with UV radiation (Bernhard et al., 2023). $O_3$ is a strong oxidant and can react with other trace
gases, including reduced sulfur compounds such as dimethyl sulfide (DMS) and carbon disulfide ($CS_2$) (Andreae et al., 1985; Hofmann et al., 1992; Persson and Leck, 1994). Moreover, the oxidation of DMS and $CS_2$, indirect precursors of COS, was reported as a potential bias of tropospheric COS measurements obtained after cryogenic sampling (Hofmann et al., 1992). On top of this, the amount of COS was found to be lower if sampled in presence of $O_3$ (Engel and Schmidt, 1994). Therefore, a number of oxidant removal substances, such as manganese dioxide ($MnO_2$), cotton wadding, sodium carbonate ($Na_2CO_3$),
potassium hydroxide (KOH) or a KI/glycerol/Vitex solution have been deployed to remove oxidants during cryogenic sampling, in both tropospheric and stratospheric applications (Andreae et al., 1985; Engel and Schmidt, 1994; Hofmann et al., 1992; Persson and Leck, 1994 and references therein). Among these, scrubbers based on cotton wadding were tested and proved to be effective for tropospheric cryogenic samples (Hofmann et al., 1992; Persson and Leck, 1994). However, only $MnO_2$ has been deployed for stratospheric applications (Engel and Schmidt, 1994). Therefore, we tested multiple substances
for their $O_3$ scrubbing efficiency and their influence on the mole fractions of the analysed gases. During the first laboratory tests, we found that $MnO_2$ interacted strongly with multiple tracers. In order to find a suitable $O_3$ scrubber that would perform well under stratospheric conditions, we conducted a series of experiments, as described below.

We designed an experimental setup (Figure A1) that obtains an air mixture with high $O_3$ at low pressure and low temperature,
to assess the performance of $O_3$ scrubbers under stratospheric air conditions. Air from a laboratory-made synthetic air (~79% $N_2$, ~21% $O_2$) cylinder containing mole fractions of the measurable tracers, close to stratospheric conditions (1435.7 ppb $CH_4$, 0 ppb $N_2O$, 392.67 ppm $CO_2$, 0 ppt COS, and 0 ppb CO), was flowed through a custom-made Y-shaped quartz tube, which passed through a UV lamp that served as an $O_3$ generator. With a QCLS-controlled mass flow of 50 mL min$^{-1}$ and a path length of about 6 inches (~ 15.2 cm) through UV radiation, it was possible to generate up to 3500 – 4000 ppb $O_3$. The air could be
directed to each side alternately using a three-way valve, allowing measurements of the air after $O_3$ generation, with or without an oxidant scrubber. The tube passed through a polystyrene box, where 193 K freeze packs could be inserted to simulate low- to mid-stratospheric temperatures (air temperature could reach as low as roughly 213 K). The flow and pressure of the air in the sampling lines were controlled by the QCLS frontend, with the mass flow typically maintained at 50 mL min$^{-1}$ and the pressure reaching as low as 250 hPa. A bypass channel directly connected to the QCLS was also available for reference
measurements.



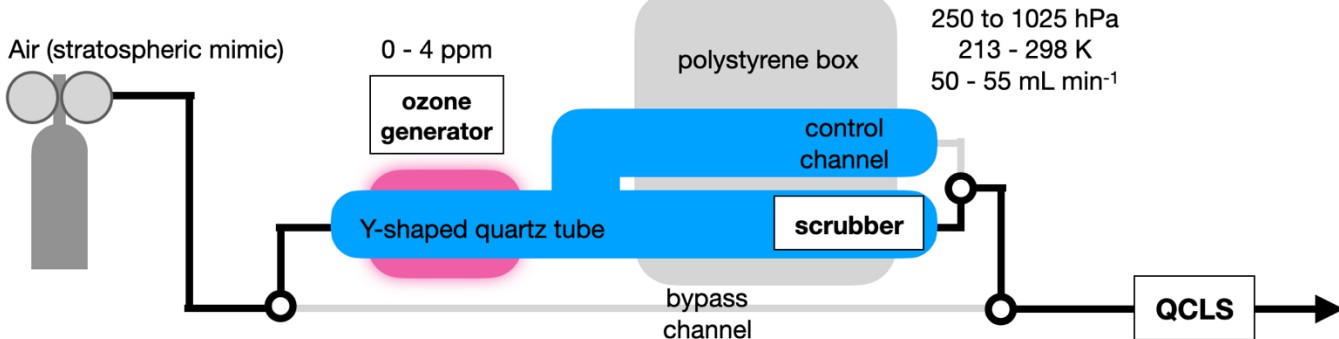

**Figure A1: the experimental setup for the O$_3$ scrubber testing. The black circles represent three-way valves. Apart from the quartz tube, all connections were realised with stainless steel tubing.**

The initial tests focused only on the performance of O$_3$ scrubbers based on cotton, supported by available research on the consumption of O$_3$ by contact with fabrics (Andreae et al., 1993; Coleman et al., 2008; Hofmann et al., 1992; Persson and Leck, 1994). Surgical cotton wool (approximately 3 - 10 g of cotton) was inserted on one side of the Y-shaped quartz tube. While cotton scrubbers seemed to work initially (Table A1), in later tests it was noticed that cotton would quickly lose efficiency, in particular after some storage time. While "new" cotton pads still exhibited some O$_3$ scrubbing capacity, the same

pads, after storage ("old" cotton), showed almost no scrubbing ability.

It was observed that the efficiency of O$_3$ scrubbing changed significantly depending on whether nitrile gloves were used or not while handling the cotton pads. This led to the hypothesis that O$_3$ was mostly removed by reaction with skin oils rather than with the cotton itself. This hypothesis was further supported by existing literature (Coffaro and Weisel, 2022; Coleman et al.,

2008; Zhou et al., 2016). Following Coffaro and Weisel (2022), knowing that squalene, a triterpene, accounts for 12% of skin oils composition (Picardo et al., 2009), squalene-based scrubbers were also tested. Consequently, the potential impacts of the reaction between O$_3$ and squalene on other trace gases were investigated. The squalene-based O$_3$ scrubbers consisted of one drop of laboratory-quality squalene (Sigma-Aldrich, ≥ 98%) deposited with a Pasteur pipette on glass wool, which had been previously proven inert to the analysed trace gases.


We designed the final experimental setup with five possible configurations:

  a. No O$_3$ generation, air flowed through the control channel
  b. O$_3$ generation, air flowed through the control channel
  c. No O$_3$ generation, air flowed through the cotton/squalene scrubber
d. O$_3$ generation, air flowed through the cotton/squalene scrubber
  e. Total bypass channel





The data from the experimental time series was selected and designated to the respective configuration. These configurations were then used as categorizations to perform an ANOVA test, which would eventually corroborate significant differences

between each species' mole fraction, depending on the experimental configuration.

The results are presented and discussed in the following paragraphs. The possible coincidental removal of COS by scrubbing substances is just briefly assessed and is presented in Sect. S1 in the Supplement. The ANOVA test results on the effect of the squalene scrubbers on other tracers are reported in Sect. S2 in the Supplement.

**Results of tests on $O_3$ scrubbers**

As reported in Table A1, the initial tests on cotton $O_3$ scrubbers (performed in 2023) resulted in the quantitative removal of about 3.5 ppm $O_3$, lasting up to roughly 2.5 to 40 L of air with a flow rate of 50 mL min$^{-1}$, depending on the amount of cotton used. However, months later, cotton pads from the same bag lost their scrubbing capacity after only about 0.05 to 0.75 L of air, soon showing a clear drop in performance and allowing progressively more $O_3$ to pass through. As explained in Sect. 2.3,

it was observed that the efficiency of $O_3$ removal was significantly higher when the cotton was handled with bare hands and eventually led to squalene-based scrubbers.

We found that one drop of squalene from a Pasteur pipette on glass wool was sufficient to quantitatively remove $O_3$ up to roughly 12 L of air without showing any sign of $O_3$ breakthrough (and we speculate it could have possibly scrubbed for an

even longer duration). When no $O_3$ was generated, no significant differences were observed for COS, indicating that nor cotton, nor squalene caused COS contaminations when interacting with the air samples.

**Table A1: summary of the first experiments regarding O3 scrubbers, which eventually led to the choice of focusing on squalene-**
**based scrubbers for the laboratory tests.**

| Scrubber type | Capacity of scrubbing ~3.5 ppm $O_3$ (air volume) | Influence on $O_3$ | Influence on COS |
|---|---|---|---|
| "Old" cotton, handled with gloves (1 cotton pad, ~ 0.3 g of cotton) | ~ 50 – 750 mL | Quantitative removal at the beginning, followed by progressive $O_3$ increase | Not significant, as long as the scrubber worked |
| "Old" cotton, handled with bare hands | ≥ 5000 mL | Quantitative removal | Not significant |



| | | | |
|---|---|---|---|
| (1 cotton pad, ~ 0.3 g of cotton) | | | |
| "New" cotton, handled with gloves (1 cotton pad, ~ 0.3 g of cotton) | ~ 800 - 2500 mL | Quantitative removal at the beginning, followed by progressive $O_3$ increase | Not significant, as long as the scrubber worked |
| Squalene (1 Pasteur-pipette drop, ~ 0.05 mL, on rock wool) | ≥ 12000 mL | Quantitative removal | Not significant |
| No scrubber | - | ~3.5 ppm measured | 0 – 40 ppt increase |







**Figure A2: ANOVA test representation for COS (panel a, top) and O3 (panel b, bottom). The red lines represent the median, while the edges of the blue boxes correspond to the 25th (bottom) and 75th (top) percentile. Black whiskers extend to most extreme data point.**


The results of the ANOVA test on squalene-based scrubbers are represented in Figure A2 for both COS and $O_3$. The corresponding p-values are reported in Table S7 and S8 in the Supplement. It is clear that squalene removes $O_3$ quantitatively: when in place, $O_3$ mole fraction shows no significant difference between the scrubbing squalene and the configurations where

$O_3$ is not generated (p-value > 0.05).

Regarding COS, significant differences (p-value < 0.05) were found between the configuration where $O_3$ flows through without any scrubber and all other configuration. However, it is noteworthy that significant differences in COS mole fraction were also observed between the configuration where $O_3$ is generated and then removed by squalene, and the bypass configuration. The

air mixture used for this experiment was prepared in the laboratory and contained 0 ppt COS. The possible reasons behind these results are discussed in the following paragraph.

The effect of squalene on $CO_2$, CO, $N_2O$ and $CH_4$ was tested just briefly and is shown in Fig. S3-S6 and Tables S3-S6 in Sect. S2 of the Supplement. Overall, CO trends resembled very closely the ones observed for COS, while no significant variance

was observed for all other tracers.

**Discussion over the results of tests on $O_3$ scrubbers**

Our study showed that cotton had only limited $O_3$ destruction efficiency over time (Table A1),in spite of its employment as an $O_3$ scrubber for reduced sulfur compounds in previous studies (Andreae et al., 1985; Hofmann et al., 1992; Persson and Leck, 1994). Existing literature reports reactions of cellulose (the primary constituent of cotton) with $O_3$ leading to the

formation of carbonyl and carboxyl groups on cellulose itself (Valls et al., 2022; Zhang et al., 2024). We speculate that the high abundance of $O_3$ (up to about 3500 ppb) may have saturated the cotton rapidly. This concentration is comparable to that of the stratospheric $O_3$ layer (Ansmann et al., 2022). Moreover, when the cotton was used several months after the first opening of its package, it is possible that its exposure to atmospheric $O_3$ (or other oxidants) may have compromised its performance.

As reported in the previous paragraph, squalene was confirmed to be an effective and efficient $O_3$ scrubbing substance, as we expected after consulting existing literature (Coffaro and Weisel, 2022; Coleman et al., 2008; Zhou et al., 2016). Squalene scrubbed $O_3$ in laboratory tests even down to 250 hPa and around 213 K. Unfortunately, since its testing began after the presented campaigns, squalene-based scrubbers have never been deployed in actual fieldwork.



As shown in Figure A2, COS mole fraction was found significantly higher than the bypass channel when $O_3$ was generated for both with and without the squalene scrubber. Assuming the glass to be inert, this observation implies that COS is produced either (i) at the UV lamp, likely from traces of VOC or other impurities in the supply gases or (ii) in the tubing downstream of the quartz glass by reactions between $O_3$ and wall contaminations. The COS mole fraction was significantly lower when ozonated air was measured after passing through the squalene scrubber (or through the cotton scrubber, before saturation) than

when it was measured after the control channel. We speculate that new volatile compounds containing carbonyl groups, possibly impurities in the compressed air mixture used for the experiment and/or products of reactions between impurities and $O_3$, may have influenced the QCLS spectrum, biasing COS measurements. Different studies report the creation of carbonyl and carboxyl groups on both saturated and unsaturated carbon polymers (Cataldo, 2001; Valls et al., 2022; Zhang et al., 2024; Zhou et al., 2016). However, these reaction products should not contain sulfur compounds. Nevertheless, squalene reduced

this bias when compared to the configuration without $O_3$ scrubbing. Therefore, we believe that most of this bias could have been produced either by products of photochemical reactions due to the UV-lamp employed for $O_3$ generation, or by reactions of $O_3$ with impurities or with experimental components. In particular, a phenomenon described as "ozone cracking" is known to affect polymers, such as (vulcanised) rubbers (Cataldo, 2001; Salomon and Van Bloois, 1963; Tse, 2007) and polymers which may be containing sulfur. Therefore, we also speculate that the reaction of $O_3$ with squalene and its consequent removal

may have mitigated these reactions, reducing the production of COS during the performed analyses. Overall, there is no clear indication that squalene would negatively bias COS measurements, while the presence of $O_3$ could be detrimental for the measurements, in particular when in presence of unsaturated polymers, which may be prone to degradation. Unfortunately, the effect of $O_3$ on long-term stored samples was not investigated within this study.

**Data availability**

The data used in this work are available from https://doi.org/10.5281/zenodo.15749915 (Zanchetta et al., 2025).

**Author contribution**

HC and MK conceived the concept, SvH, AZ and HC designed the experiment, AZ, SvH, JH, RK, TL, MR, ML, PN, SLB,

and HC collected the data. AZ, SvH, and HC wrote the manuscript with contribution from all authors.

**Competing interests**

Huilin Chen is a member of the editorial board of Atmospheric Measurement Techniques.

**Acknowledgements**

We are grateful for the support during the preparation of the campaigns by Bert Kers, Marcel de Vries and Marc Bleeker at the Center of Isotope Research. We would like to thank the colleagues who collaborated during the campaigns, especially



Maria Elena Popa, Johannes Laube and Johannes Degen. The AirCore flights were partially supported by the ESA project FRM4GHG.


**Financial support**

This research was supported by the ERC advanced funding scheme (AdG 2016 project no. 742798, project abbreviation COS-OCS), and by the Ruisdael Observatory infrastructure cofinanced by the Dutch Research Council (NWO, Grant No. 184.034.015) and ICOS Netherlands. This work was also supported by the Natural Science Foundation of China (42475115) and by the Fundamental Research Funds for the Central Universities (14380235).

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
