# Peer review of "Balloon-borne Stratospheric Vertical Profiling of Carbonyl Sulfide and Evaluation of Ozone Scrubbers"

_EGUsphere, 2025_

## Author Comment (AC1)

**Summary**

This manuscript evaluates strategies to mitigate ozone ($O_3$) contamination during stratospheric sampling of carbonyl sulfide (OCS) using the AirCore balloon system. The primary finding is that $O_3$ does not appear to significantly affect OCS measurements across the atmospheric column: AirCore profiles with and without $O_3$ scrubbing agree well with independent datasets from ACE-FTS and SPIRALE. Observed differences from these datasets fall within the QCLS measurement precision and the expected natural variability, particularly in high-latitude regions where seasonal OCS changes can reach ~150 ppt (https://gml.noaa.gov/hats/gases/OCS.html).

The authors tested three inlet $O_3$ filtration strategies: cotton, magnesium perchlorate [$Mg(ClO_4)_2$], and squalene. Laboratory results show that cotton is ineffective at removing $O_3$ (Figure S2), and squalene was not deployed during balloon flights. Thus, only $Mg(ClO_4)_2$-filtered and unfiltered ("free inlet") profiles provide field-relevant comparisons. However, because the $Mg(ClO_4)_2$ dryer was not characterized in laboratory experiments, its $O_3$ removal efficiency in the field remains uncertain.

The motivation for $O_3$ mitigation stems from Engel et al. (1994), which reported up to 50% OCS loss when co-sampled with stratospheric $O_3$ in balloon-borne cryogenic samplers. Notably, in the nearly three decades since, no widely cited studies have confirmed such losses, suggesting that the effect may be specific to cryogenic sampling techniques. If so, the null result reported in this manuscript has important implications for interpreting historical in-situ stratospheric trace gas records.

*Authors' response (AR): the authors would like to thank the referee for the generally positive comments and the provided feedback. Our responses will be organized question-by-question in paragraphs formatted similarly to the present one. **Major modifications** in the preprint will be presented as **italic bold text** together with their respective page and line numbers, when applicable.*

**Major Comments**

1. **Laboratory results vs. Engel (1994)**

The laboratory results presented in this manuscript appear to contradict the findings of Engel et al. (1994). Specifically, Figure A2 shows a slight enhancement in OCS (~40 ppt) in the presence of stratospheric $O_3$ concentrations, whereas Engel reported up to a 50% loss of OCS under similar $O_3$ levels. Given the significance of these laboratory results to the paper's conclusions, I recommend moving most of the information currently in Appendix A into the main body of the manuscript. However, it should be noted that the enhancement reported here is less than 10% of typical ambient OCS levels and approaches the instrument's stated precision (~25 ppt), making it unclear whether the observed difference is statistically meaningful.

*AR: thank you for the positive opinion about these findings and for the suggestion. We agree that the information reported in the Appendix is also scientifically significant.*

*Firstly, we would like to stress a significant difference between laboratory tests and real stratospheric conditions. Engel et al. (1994) reported up to a 50% loss of OCS, which is in contrast to our test results as stated in the Discussion section of the Appendix (page 31, lines 600 – 604),* "COS mole fraction was found significantly higher than the bypass channel when $O_3$ was generated for both with and without the squalene scrubber. Assuming the glass to be inert, this observation implies that COS is produced either (i) at the UV lamp, likely from traces of VOC or other impurities in the supply gases or (ii) in the tubing downstream of the quartz glass by reactions between $O_3$ and wall contaminations". *In other words, although a COS enhancement is noticeable when $O_3$ is generated in laboratory conditions, this is likely due to either the gas or the gear employed during the experiment and therefore unlikely to occur during stratospheric sampling.*

*We understand the suggestion of moving (part of) the $O_3$ scrubbers experiments from the Appendix to the main text of the manuscript. However, the choice of reporting laboratory experiments as a whole in the Appendix was made due to the fact that different scrubbers were deployed in field campaigns and in the laboratory tests. In particular, the most efficient $O_3$ scrubbers (squalene) were only tested in laboratory conditions. Overall, we believe that this difference led to distinct aspects of a single study and have chosen to report laboratory findings in a standalone section. We believe this shall also improve the readability of the manuscript as a whole.*

2. **Appropriateness of multi-year dataset comparisons**

Due to the substantial natural variability in tropospheric OCS abundance and in stratospheric OCS driven by transport processes, I do not consider it appropriate to perform a quantitative comparison of short-duration datasets separated by several years. I therefore recommend removing Figures 2, 6, and 7 as well as Tables 3 and 5. Table 2 could be moved to the Supplementary Information or omitted entirely, as its contents are already conveyed in Figure 1. Alternatively, a comparison with $N_2O$ or other stratospherically photolytic tracers such as CFCs could allow for quantification of stratospheric transport, but this would require substantial additional analysis and constitute a major revision to the manuscript.

*AR: thank you for this insightful feedback. We would like to clarify some aspects that may have not been explained thoroughly enough in the manuscript.*

*Figure 2 shows COS profiles and discrete samples from all campaigns, plotted against altitude above the tropopause. The figure includes profiles from all campaigns but is not intended for quantitative comparison between them. Instead, it shows the altitudinal trends of COS due to the stratospheric sink at different latitudes. In Sect. 4.1.3, page 14, lines 314-319, the following description is given* "Figure 2 shows the measured COS AirCore profiles and LISA samples from all campaigns, plotted against altitude above

tropopause (see Sect. S3 in the Supplement). TRN1 is not presented in this figure, since the tropopause height could not be estimated due to missing temperature and relative humidity. Differences of up to ~100 ppt can be clearly seen between the measured profiles. However, these differences are not constant with altitude and do not show any clear trend over the time span of the campaigns. Moreover, these differences do not show any clear relationship with the different inlets employed". *We believe the observed profiles above the tropopause reflect different stratospheric loss rates with altitude and latitude and are worth showing.*

*Table 3 reports the differences between measurements obtained with different samplers (AirCore against LISA or BigLISA) on the same flights. Therefore, the quantitative comparison has been realized between short-duration datasets obtained over the same vertical column. This was done as altitude mapping, which is a known challenge for AirCores (Membrive et al., 2017; Tans, 2022; Wagenhäuser et al., 2021) and (Big)LISA should have provided a mean to verify vertical profiles obtained from AirCore sampling. Although BigLISA samples resulted to be outliers for COS, we believe the findings related to this instrument are still valuable to improve COS sampling with LISA and, more broadly, with MLF bags and were therefore included in the manuscript.*

*Table 5 and the associated Figure 6 and Figure 7 report a quantitative comparison between the short-duration datasets collected and measured in the reported campaigns and the average of numerous ACE-FTS profiles over several years. The objective of this comparison was to quantify possible biases between the retrieved stratospheric COS and ACE-FTS observations and investigate trends and daily features of stratospheric COS. This analysis was performed following previous studies (Hannigan et al., 2022; Krysztofiak et al., 2015; Velazco et al., 2011), which used ACE-FTS data averaged over several years to investigate agreements and differences with their datasets. In this study, we also presented linear regressions as a tool to quantify biases more directly. Moreover, several studies reported no significant trends in stratospheric COS (Barkley et al., 2008; Coffey & Hannigan, 2010; Rinsland et al., 2008; Toon et al., 2018). Therefore, we consider this analysis scientifically significant and opt to keep it in the manuscript.*

***Table 2 was moved to the Supplementary information in a new Section, numbered S1. The other Sections in the Supplement and the Tables in the main text were numbered again, accordingly.***

*A comparison of $CH_4$ with $N_2O$ has been presented in Sect. S4 (now S5) in the Supplement as a tool to quantify atmospheric transport. Moreover, we investigated the relationship between COS and $N_2O$. A detailed analysis of this relationship will be presented in a manuscript that is being prepared for submission.*

3. **Concerns with LISA and BIG LISA "bag" data**

The manuscript includes significant caveats regarding the LISA and BIG LISA MLF "bag" sampler data. In Section 4.1.2, the authors note that the 30229-U MLF bags used in

BIG LISA have a manufacturer warning: "Although the deployed bags are indicated as suitable for sulfur compounds, they are not recommended for low-ppm volatile organic compounds due to background levels" (Sigma Aldrich, 2025). A similar warning appears on the Sigma Aldrich website for the 30228-U bags used in LISA: "Not recommended for low ppm VOCs due to background levels (we recommend the SupelInert PVDF Tedlar alternative film for VOCs)". Considering that a common VOC such as $CS_2$ can oxidize to OCS with ~80% efficiency, even sub-ppm VOC contamination could significantly bias atmospheric OCS measurements (<1 ppb). In addition, a subset of the LISA OCS data appears to have been omitted by the authors without explanation (see Section 2.1.2). Given these concerns, I recommend removing the LISA dataset analysis from the manuscript.

*AR: thank you for pointing out the caveats. Despite the manufacture's warnings, we demonstrated in the laboratory that reliable COS measurements using such bags are achievable. This effort piggybacks on existing measurements of $CO_2$, $CH_4$, and CO using these bags (Hooghiem et al., 2018). We agree that careful further work is needed to achieve high accuracy COS measurements. Nevertheless, we believe some of our initial LISA results are worth reporting, as they may be valuable for similar and future studies.*

*The exclusion of some LISA samples was explained at page 5, lines 149-151: "The leftover volume of one of these samples was insufficient for analysis (SOD3 – L4), while two others showed unusually high mole fractions for several of the analysed gas species (SOD2 – L4, SOD5 – L3). These three samples were labelled as outliers and will not be presented in this work, and are not included in Table 2". We believe the stated reasons to be a reasonable justification for labelling the aforementioned samples as outliers and exclude them from the following analysis.*

*Following your advice, we have significantly modified the presentation of the LISA results. The following changes were implemented in the main text:*

*Page 5, line 149-150: …while two others showed unusually high mole fractions for several of the analysed gas species (SOD2 – L4, SOD5 – L3)***, in spite of the pre-conditioning of these bags before flight***.*

*Page 5, line 151: These three samples were labelled as outliers and will not be presented in this work**.*

*Pages 13-14, lines 296-306: "[…] Unfortunately, it has not been possible to assess the cause of this contamination precisely. Given these circumstances and the impossibility of applying any correction to these results, BigLISA will be left out of the discussion and comparisons with other datasets. However, as described in Sect. 2.1.2, during the SOD campaign we introduced a pre-treatment technique****, based on previous laboratory tests,*** that has **mitigated** this issue for COS. Filling and vacuuming the bags with a stratospheric-mimicking gas seemed to have reduced the contamination significantly for most LISA samples, as previously reported in Fig. 1 and*

Table 2. ***However, in some cases and in particular for smaller samples collected at higher altitudes (SOD2 – L4, SOD5 – L3), this pre-conditioning was not sufficient to prevent biases that are likely ascribable to the bags employed during the campaign. For the remaining samples, when LISA flew on the same balloon as one of our AirCores [...]***".

*Page 13, line 299 (now line 307):* "Although **all** the deployed bags are indicated as suitable for sulfur compounds, they are not recommended for low-ppm volatile organic compounds due to background levels (Sigma Aldrich, 2025)."

**Specific Comments:**

Line 48: Remove: "Although the debate has not been fully resolved"

*AR: done.*

Line 59: Provide a reference for the QCLS

*AR: added Kooijmans et al. (2016); Stimler et al. (2009).*

Line 66:  Please explain why the impact of air samples of COS may be significant.  I can find no other examples aside from the Engel 1994 paper.

*AR: included the following sentence:* ***$O_3$, a strong oxidant, may in fact react with reduced sulfur compounds causing a variable, yet possibly significant reduction in their abundance (Andreae et al., 1990; Engel & Schmidt, 1994; Hofmann et al., 1992; Persson & Leck, 1994)***.

Line 72:  Add a reference to the Schmidt et al 2024:

Schmidt, Matthew, et al. "Trends in atmospheric composition between 2004–2023 using version 5 ACE-FTS data." *Journal of Quantitative Spectroscopy and Radiative Transfer* 325 (2024): 109088.

*AR: thank you for this suggestion. The reference has been added.*

Line 83:  Explain how Aircore and Sulfinert specifically differs from the stratospheric Cryogenic air sampler apparatus.

*AR: added "*When used to retrieve vertical profiles, AirCore sampling is realised passively ***and continunously*** along the coil ***at ambient temperature*** (Karion et al., 2010; Membrive et al., 2017; Wagenhäuser et al., 2021), ***differently from the discrete samples that can be collected by cryosamplers which, however, need to be cooled down to 27 K with liquid neon before flight (Laube et al., 2010; Schmidt et al., 1987)***.

Line 102:  Do you expect any Aqueous reactions with ozone to occur on the wetted cotton?

**AR:** *it is possible that water itself may have reacted with O₃, while for COS the scarce solubility, low temperatures during sampling and the rapid inflow through the inlet may suggest that these reactions, if occurred, should have had a marginal influence especially when compared to dilution and matrix effects.*

Line 146: What is the duration of the samples storage in: 1. MLF bags, 2. glass cylinders before analysis

**AR:** *as reported in lines 147-149 "*Here we present the analysis results of the air samples left in the sampling bags, when present, directly after the sample transfer from the MLF bags to glass flasks (these latter were not analysed on the QCLS)*". The storage in MLF bags was the period from the collection of the samples during ascent until they were analysed or transferred to glass flasks in the laboratory, which varied from flight to flight and ranged approximately between 2.5 and 3.5 hours. Glass flasks (cylinders) were not analysed on the QCLS and are therefore not part of this study.*

Lines 149-151: Without a specific reason to disregard a subset of samples, you must either present all of the data or none of it.

**AR:** *regarding SOD3 – L4, the sample left in the bag after transfer into the glass flask was simply too small for analysis (it was collected and stored for different analyses, but was not measured on the QCLS). The other two samples (SOD2 – L4, SOD5 – L3) were excluded due to very clear signs of contamination for all tracers, possibly due to tropospheric air mixing with the sample during the transfer of these samples in the glass flasks. We thought it would have been not meaningful to present these samples as, contrarily to the BigLISA ones, they were clear outliers for all the measured tracers and not just for COS – a strong indicator of a general contamination of the sample and not of a specific problem related to COS sampling.*

Lines 205-207: Please present the data for the QCLS precision.

Lines 211-212: Present the data or references for the QCLS precision.

**AR**: *lines 211-212 about the QCLS precision were moved to lines 208-209. References of studies that used the same instrument have been added* (Tong et al., 2023; Vinković et al., 2022)*. Allan deviation plots of the 5 gases measured by the Aerodyne QCLS during AirCore campaigns in 2018, 2023 and 2025 are shown in Figure 1. Note that the y-axis is not on a logarithmic scale. Source data for these plots are, for each campaign, an arbitrarily selected brief period (1-3 hours) of measurements of a single working standard, without any calibrations performed. The selected data and results represent the quality attained during typical campaign operations (e.g., after instrument transport and with exposure to variable ambient temperatures) and are inferior to what may be attained in a carefully controlled laboratory setting. Note that the noise floor is typically observed already after ~100 seconds, after which instrumental drift becomes an appreciable factor. Given that calibration was performed shortly before and after profile measurement (using*

*3-minute-long averages of working standards), actual profile quality may be expected to be less impacted by instrumental drift.*

[Figure]

*Figure 1: Allan deviation plots of the 5 gases measured by the Aerodyne QCLS during AirCore campaigns in 2018, 2023 and 2025.*

Line 215:  Which version of ACE-FTS are you using v5.2? v5.3?

**AR:** *we apologise for having overlooked this specification. We are using v5.3 and it has now been specified in the manuscript and references to Boone et al. (2023) and Schmidt et al. (2024) have been included there as well.*

Line 216:  Add Boone et al 2023. And Schmidt et al 2024:

Boone, C. D., P. F. Bernath, and M. Lecours. "Version 5 retrievals for ACE-FTS and ACE-imagers." *Journal of Quantitative Spectroscopy and Radiative Transfer* 310 (2023): 108749.

*AR: done (see above).*

Line 228: Do the 502 and 1681 profiles correspond to global samples at those latitude bands or localized profiles over TRN and KRN?

*AR: the latitude bands were chosen at ± 2° from the sampling locations, to select localized profiles over TRN and KRN, similarly to what was done in Krysztofiak et al. (2015).*

Line 244: This analysis would be more quantitative if you compared with the Age of Air parameter or another photolytic species like N2O.

*AR: we understand this remark and agree with the referee that other parameters would provide a more precise comparison than altitude itself. Here, we used altitude so that the results can be comparable to those of previous studies (Glatthor et al., 2017; Krysztofiak et al., 2015; Leung et al., 2002), since altitude was the only available parameter to realise a COS comparison. Citing Krysztofiak et al. (2015), for instance: "We observe for polar latitude a decrease of OCS vmr with altitude, from 420±100 pptv for altitudes lower than 17 km and then a decrease from 460 pptv to 150 pptv at 22 km. During the STRAT campaign in July 1996, the MkIV instrument observed similar behaviors with OCS constant vmr values of 440 pptv below 14 km (Leung et al., 2002) and then a decrease to 120 pptv at 22 km". The following sentence has been added at line 281 in Sect. 4.1.1 of the Discussion: "This is consistent with observations reported in previous studies at comparable latitudes (Leung et al., 2002; Toon et al., 2018). **Although altitude itself does not provide a precise proxy for quantitative comparisons of vertical profiles, this agreement is a strong suggestion of robust stratospheric COS measurements**".*

*An evaluation of altitude mapping and possible transport effects was realised using the relationship between $CH_4$ and $N_2O$ (Sect. S5 in the Supplement). A detailed analysis of the relationship between the measured stratospheric COS and $N_2O$ is being prepared for a future publication.*

Line 263: The MkIV spectrometer utilized by Toon et al is balloon borne into the stratosphere, but this is a long path solar FTIR measurements so it may be more appropriate to include with the remote sensing measurements.

*AR: the citation has now been placed with the ones regarding remote sensing measurements.*

Line 278: Please provide more details for the "direct reaction of other gas species with O2."

*AR: the sentence was modified to "In the case of SOD3, another possibility could be direct **oxidation reactions** of **reduced sulfur** gas species with $O_2$."*

Line 284: How does your observed tropospheric variability compare with NOAA GML flask network measurements for high latitudes?

*AR: a quick look at NOAA's flask data from Arctic stations shows reasonable agreement with the observed variability (the high-latitude campaigns were performed around decimal years 2021.6 and 2023.6). In August 2021, NOAA's flask data ranged from approximately 342 to 453 ppt, while in August 2023, the COS tropospheric molar fraction ranged from about 361 to 436 ppt (see Fig. 2). When plotting the differences in COS between consecutive measurements (Fig. 3), we find variations of up to 200 ppt (ranging from -150 to 50 ppt) over periods as short as one week (corresponding to a decimal time of ~ 0.019). Overall, we believe these findings warrant a separate investigation that could include wind directions and trajectory analysis. Nonetheless, the preliminary analysis seems to support the tropospheric variability observed during our campaigns.*

[Figure]

*Figure 2: NOAA's tropospheric COS flask measurements at arctic stations for the period covered by the field campaigns of this study.*

[Figure]

*Figure 3: COS differences between consecutive measurements of NOAA flasks.*

Line 290: How is the variable lapse rate an indicator of OCS convective transport? Have you preformed trajectory analysis?

*AR: following WMO's definition, the thermal tropopause is defined as the lowest level where the atmospheric lapse rate decreases to 2 K km$^{-1}$ and remains below this value for at least 2 km. For SOD1 and SOD5 (and in particular for this latter) this condition is not strictly respected. Moreover, in SOD5 it is possible to observe a less steep decrease in water vapour than in the previous flights, with water being registered by the radiosonde up to roughly 17 km, providing a strong suggestion of convective transport from the troposphere. Unfortunately, trajectory analyses were not performed.*

Line 299-301: The low-ppm VOC warning is listed for both LISA and Big-LISA MLF bags.

*AR: corrected (see above).*

Line 317: The absolute differences in OCS above the tropopause are not useful. You must compare with N2O to understand stratospheric transport and age of air factors, and correct for N2O increasing trend.

*AR: we agree with this comment. However, as reported in previous ARs, this paragraph was meant to be a qualitative description of observed trends and features of stratospheric COS. As stated earlier in this reply, the fact that altitude itself does not provide a precise mean for comparing stratospheric features has been specified in the manuscript. However, previous studies used differences over altitude to describe the features of COS profiles (Glatthor et al., 2017; Krysztofiak et al., 2015; Leung et al., 2002). The authors believe Sect.*

*4.1.3 provides an overview of the reasons behind the differences observed in the measured profiles and, although not quantitatively, covers thoroughly the possible causes of these discrepancies. A comparison with $N_2O$ was presented in the Supplement for $CH_4$ and, as already mentioned above, a thorough analysis of the relationships between COS and $N_2O$ during these campaigns has been realised and will be presented in a manuscript that is being prepared for submission.*

Line 323:  Include Schmidt et al 2024 to for stratospheric OCS trends.

**AR:** *done.*

Line 331:  On line 327 the authors state that no quantification of sample loss is available, but here they claim that "differences remain marginal."  Please attempt to provide more quantification of the uncertainties

**AR:** *given the AirCore's design, and assuming the sampler is built and sealed properly, sample loss can only occur at the open end of the coil. The outflow begins with the most recently sampled air, namely, the portion closest to the ground level. Each aliquot of air in the tropospheric part of the profile corresponds to a much smaller altitude difference than on its stratospheric counterpart (see Sect. 2.2, lines 209-211). Therefore, although the sample loss could not be quantified, it would have only slightly affected the overall altitude mapping and in particular the affected portion would most likely have mixed with the push gas during analysis.*

*With regard to the other mentioned sources of uncertainty, namely dead volumes of tropospheric air or fill gas, impurities in the scrubbers or effects due to instrumental components, it is quite complex to provide a proper estimate of their contribution to the overall uncertainty. Dead volumes, if present, are unknown but may account up to very few $cm^3$ of air over samples ranging between 800 – 1600 $cm^3$. The volumes of the impurities that may reside in cavities within the deployed scrubbers should be even smaller, together with the effects of the instrumental components. Even if hard to quantify, the uncertainties related to these aspects should remain negligible when compared to differences due to atmospheric transport and day-to-day variability (see, for instance, the estimated tropospheric variability). We regret not being able to give a more precise estimate of these uncertainties, which remain a known challenge of the AirCore sampling technique (Karion et al., 2010; Membrive et al., 2017; Tans, 2022).*

*The following sentence has been now included in the main text (Lines 340 -346): "*These may include mixing with dead volumes of tropospheric air or fill gas, impurities in the deployed scrubbers or effects due to the instrumental components (e.g., O-rings, tubing). **However, even if these uncertainties are hard to quantify, they may account for up to a few $cm^3$, against collected sample volumes ranging roughly between 800 – 1600 $cm^3$.** Therefore, we assume that differences due to instrumental effects remain marginal, while we believe that the day-to-day variability and long-term trends in COS mole fractions are the most important cause for the observed differences."

Line 339: The changing relationship between Methane and N2O is interesting and either points to major instrument issues or more complex stratospheric dynamics and should be discussed more here.

*AR: we agree that this relationship is relevant as it is symptomatic of something unusual within the discussed profiles. As explained in Sect. 2.1.1 (lines 112-119), the deviation found in the $CH_4$-$N_2O$ relationship can be ascribed to the design of the double-sided AirCore. In this instrument, since the remaining portion of fill gas sits in the middle of the coil instead of being pushed at one of the ends during sampling – which is typically the end at which the measurement starts. This gives it more time to mix with the highest side of the samples during analysis and, since it must travel through the entire coil before reaching the analyser, this portion likely experiences a stronger smearing effect than in the standard design.*

Line 370: The temporal differences really limit any kind of quantitative analysis.

*AR: we understand the concerns in this regard. However, as reported earlier, this analysis has been performed to make it comparable to previous studies (Glatthor et al., 2017; Hannigan et al., 2022; Krysztofiak et al., 2015; Velazco et al., 2011), although mostly focusing on comparisons with remote sensing observations. Although the dataset of Krysztofiak et al. (2015) and the profiles presented in this study lie 10 to 14 years apart, the authors believe that a quantitative comparison may still be useful to investigate (stratospheric) COS trends over time and possible daily features that may have been captured in the different studies.*

Line 383: How does the 8% difference compare with the decadal change in stratospheric OCS abundance?

*AR: thank you for this important remark. Since 2009, no significant trends in stratospheric COS abundance have been reported (Barkley et al., 2008; Coffey & Hannigan, 2010; Rinsland et al., 2008; Toon et al., 2018). Therefore, it is reasonable to assume that this difference may be due to reciprocal instrumental biases or specific atmospheric conditions during sampling rather than long-term trends. This has been explicated as follows in lines 373-375: "[...] and, most importantly, between 10 and 14 years apart from each other.* ***However, it is relevant to mention that no significant trends were observed for stratospheric COS in recent years (Barkley et al., 2008; Coffey and Hannigan, 2010; Rinsland et al., 2008; Toon et al., 2018)****. To realise a meaningful comparison [...]".*

Line 400: This qualitative comparison is not useful, just refer to Figure 3

*AR: A reference to Figure 3 was included at line 408.*

Line 413: What version of ACE-FTS are you using?

*AR: the authors apologise again for overlooking this detail. Version 5.3 is now specified in the manuscript together with the references to Boone et al. (2023) and Schmidt et al. (2024).*

Line 476:  Add discussion of Squalene-based scrubbers to the main body of the text.

Line 479:  This is a major result, and you should highlight the difference from Engel's 1994 paper.

*AR: thank you for your positive opinion and the expressed interest about our findings about $O_3$ scrubbing materials. As stated in the reply to the first major comment, the observed enhancements of COS molar fraction during laboratory tests shall not be compared to the findings of Engel & Schmidt (1994) due to the inherently different air mixtures involved in the experiments. It is unclear whether COS may have been simultaneously removed by $O_3$ and produced by the reaction of $O_3$ with precursors (CS$_2$, DMS) that may have been present as impurities in the air mixture employed for the experiments. This was been specified in the Appendix, as follows:*

*Lines 606-613:* "As shown in Fig. A2, COS mole fraction was found significantly higher than the bypass channel when $O_3$ was generated for both with and without the squalene scrubber. **This is in contrast with the $O_3$-induced COS loss reported in previous studies (Andreae et al., 1990; Engel and Schmidt, 1994; Hofmann et al., 1992; Persson and Leck, 1994).** Assuming the glass to be inert, this observation implies that COS is produced either (i) at the UV lamp, likely from traces of VOC or other impurities in the supply gases or (ii) in the tubing downstream of the quartz glass by reactions between $O_3$ and wall contaminations. *Overall, it is unclear whether COS may have been simultaneously removed by $O_3$ and produced by the reaction of $O_3$ with COS precursors (e.g., CS$_2$, DMS) that may have been present as impurities in the air mixture employed for the experiments. This limits the direct comparison with stratospheric field studies (Engel and Schmidt, 1994).* Nevertheless, COS mole fraction was significantly lower when ozonated air was measured after passing through the squalene scrubber […]".

*With regard to the positioning of these findings within the manuscript, we decided to include them in the Appendix instead of the main text to improve the consistency and the readability of both sides of this study. Since $O_3$ scrubbing materials were only tested in the laboratory, and the most successful one was not deployed during field campaigns, we believe that keeping the two experiments separated – while presenting them both in the main text of the manuscript – makes the manuscript more coherent and easier to understand as a whole.*

---

## Author Comment (AC2)

The manuscript "Balloon-borne Stratospheric Vertical Profiling of Carbonyl Sulfide and Evaluation of Ozone Scrubbers" by Alessandro Zanchetta et al. compare several new methods to measure carbonyl sulfide (COS) concentrations along vertical profiles into the stratosphere using balloon-based air sampling platforms. The authors compare measured COS profiles between the different sampling instruments, from three locations and assess their results in comparison with modelled COS profiles. The methods show good overall agreement and will undoubtedly be able to provide new and insightful information to the growing science community interested in COS.

The manuscript furthermore includes an assessment of the efficiency of ozone scrubbers, based on experiments undertaken on a purpose-built setup. The authors tested cotton, a material that has been suggested some 30 years ago and found that it is not as reliable in ozone elimination as it might have been expected. They tested an alternative material in laboratory experiments, which has not yet been tested on the balloon-based air samplers, but shows very promising results.

The manuscript is very well written, the experiments are clearly described, and the overall methods are sound. In my opinion, this is a fantastic study, and the manuscript is an excellent fit for publication in AMT.

*Authors' Reply (**AR**): thank you for the very positive feedback about our study. The answers to the provided comments will be presented in italic black text and the modifications to the manuscript will be presented as **italic bold** text, citing page numbers and lines when applicable.*

There are, however, a few minor points that I suggest considering prior to publication to improve the clarity of the manuscript:

1) Instrument calibration. The observations span a period of four years, from June 2019 to August 2023. COS measurements are challenged by instability of COS in reference gas cylinders. While the authors provide information on the measurement precision, information on the long-term stability of the instrument is not presented. As this study potentially requires long-term stability of the instrument over four years, a demonstration or small assessment on the stability to underpin the robustness of the presented method would be desirable.

***AR:** we appreciate this remark and have clarified our efforts to achieve long-term stability of our observations. The long-term calibration was performed identically to previous studies about COS measurements on the same instrument (Kooijmans et al., 2016; Zanchetta et al., 2023). In addition, by using dilution of a cylinder with a known mixture of ~1 ppm COS and ~0.4% $CH_4$, we assessed our 2015-era standard in 2023 and 2024 and found no appreciable drift (<0.3%, or <2 ppt COS) over that period.*

*In 2018, our lab acquired from NOAA (Boulder, USA) a calibration standard at ~1000 times elevated and accurately known concentrations of $CH_4$, COS and CO. High-concentration cylinders are known to show much reduced drift over time than 'natural concentration' cylinders (NOAA, personal communication). This cylinder may be used to set our QCLS scale by using the accurately known $CH_4$/COS-ratio of that high-concentration standard. Pulsed admixture of a small amount of the highly concentrated gas into a trace-gas free carrier gas (either $N_2$ or an $O_2$/$N_2$/Ar mixture) allows determination of the slope (and linearity) of the COS to $CH_4$ ratio of the QCLS. After calibrating the $CH_4$ measurements to standards, we can in principle calibrate its COS measurements, infer drift in standards, or even re-assign COS values to them.*

*This assessment has been attempted in 2023 and 2024 for our 2015-era standard, which was found not to have drifted appreciably (<0.3%, or <2 ppt COS) over that period. However, currently unclear deficiencies and unknowns in our procedure (among other: repeatability, effects of zero gas, unresolved uncertainty of the COS scale of the standard; Bradley Hall, NOAA, personal communication) have so far made us hesitate to re-assign COS values to that cylinder. Tentatively, such an updated COS value may be ~ 5% (~25 ppt) lower than 10 years ago, with proportional implications to the AirCore results reported in this publication. However, in light of the uncertainty of this assessment, for this publication, we will assume that no appreciable drift has occurred in the COS of our standards.*

*The following paragraph has been added in Sect. 2.2, lines 213-219:*

**As reported in Zanchetta et al. (2023), field standard cylinders are calibrated against NOAA standards (NOAA-2004 COS scale) in the laboratory before and after each measurement period to test for drift in the molar fraction of gas species. The COS mole fraction measurements of nine cylinders are available, and five cylinders changed by less than 2.5 ppt $yr^{-1}$, two cylinders decreased by ~ 10 ppt $yr^{-1}$ and two cylinders decreased by ~ 30 ppt $yr^{-1}$. The four cylinders that drifted more than 10 ppt $yr^{-1}$ were not used as reference cylinders in the data processing. All of the cylinders were uncoated aluminium cylinders, which, according to experience at NOAA, are more prone to COS mole fraction drift than Aculife-treated aluminium cylinders. More details on the instrumental calibration, precision and stability can be found in Kooijmans et al. (2016)."**

*A dedicated, and more refined, attempt is currently being undertaken to use the high-concentration standard for calibration/monitoring of our standards. Results thereof may be reported in an upcoming publication, which may additionally allow for an assessment of the accuracy of the AirCore work presented here.*

2) Figures. Many of the figures include a lot of data. Several figures are so crowded that I find it difficult to comprehend (for example 3 a). Figures often include very small symbols (for example 3 b and c), which are too small for me to tell apart by colour, blurring the figure and the story the data tell. Other examples include Figure 1 left panel, Figure 2, Figure 3 left panel, symbol size

in Figure 3 middle and right, Figure 4, Figure 5. Can Figures 6 and 7 have larger symbols? Maybe split some of them and have Figures on single panels with full width (i.e., Figure 3 a)? Could log scales be useful to decompress the large numbers of symbols in the lower y-axis range? Or could panels b and c of Figure 3 be on top of each other as a second column in the Figure, allowing 3 a to be larger? I am not sure how to improve, but I think the present state doesn't make the most out of the fantastic data and it would be well worth to improve their clarity.

Some multi-panel Figures have letters as identifiers (i.e., Figure 3 a-c), others (i.e., Figure 1) don't, while some are referred to as second/third (p16, l377). Furthermore, the text refers to the Figures in a sequence that is different from the appearance and enumeration of the Figures, i.e., the text first refers to Figure 3, then Figure 5, then Figure 1. This might be a matter of personal taste, but greater consistency on those points would make the manuscript more accessible to me as the reader. It can be difficult to follow at times, especially when the text is jumping between different figures across different pages with focus on different events in those figures, i.e., P17, l386, where it might be useful to have markers in the figures (arrows?) to mark these events.

*AR: thank you for this feedback, and we have made the following changes in the figures.*

***The panels of Fig. 3 have been separated and Fig. 3a is now a standalone Fig. 3, while Fig, 3b and Fig. 3c are now shown as Fig. 4a and Fig. 4b, respectively**.*

*Following the referee's suggestions,* ***all multi-panels figures have been labelled with letters** (e.g., panel a, b, ...) and **the markers have been enlarged for all figures, with the exceptions of Fig. 3 (previously, Fig. 3a)**. We believe that for Fig. 3, given the presence of error bars, bigger markers would have made the figure more busy and confusing.*

*The numbering of the figures was corrected and the in-text references to figures were adapted to the labels given to the figures' panels.*

***Fig. 7 was moved to Sect. 4.2 and consequently became Fig. 5. Accordingly, Table 4 was divided in Table 4 and Table 5 reporting the SPIRALE regression results and the ACE-FTS regression results, respectively**.*

*We are grateful for the very useful and constructive comments, and believe that these changes have improved the paper's structure and generally improved its readability.*

3) Structure. The main part of the manuscript includes an Appendix, which talks about the setup to assess different O3 scrubber materials, as well as the results of experiments made using gas from one cylinder with 0 ppt COS. There are also supplementary information to the manuscript, which include more test results on O3 scrubbers using a different cylinder with a different test gas,

containing some 750 ppt COS. It is not clear to me why the O3 scrubber assessment is separated between appendix and supplementary information? It would seem to me that they could be merged into one assessment in either appendix OR supplementary information, which can then be referred to in the main text. As the assessment of the scrubbers in the appendix is made on a gas containing 0 ppt COS, I believe this doesn't allow to assess whether the scrubbers could potentially reduce COS in the sample, as it is already at 0 ppt. To me, this is an important point that could potentially have affected the outcome of the study, if tests hadn't been made with the 750 ppt COS gas as well. Here, the combined information from tests made with the 0 ppt COS and 750 ppt COS provide the robustness of the results that is needed. Therefore, the separation of those sections into Appendix and Supplements seems confusing to me and I'd suggest keep together in the manuscript. This would enable presenting the analysis in a more robust and compact way, which I think would improve the clarity of the manuscript.

*AR: Thank you for your comments on the paper structure and pinpointing on the separation of $O_3$ scrubbers between Appendix and the SI. We have made the following changes:* **the text and figures of Sect. S2 of the Supplement has been moved to the Results and Discussion sections of the Appendix.** *The tables reporting the p-values resulting from the ANOVA tests were left in the Supplement, and Sect. S2 became Sect. S3 to resemble the order in which the information is presented in the Appendix. This should also allow the reader to obtain a more thorough overview of the performed experiments and provide more detailed insights of the possible effects related to the interaction of COS, $O_3$ and the tested scrubbing materials.*

 **Specific comments:**

Title: "Ozone Scrubber Materials" instead of "Ozone Scrubbers"?

*AR: changed to "Ozone Scrubbing Materials".*

P2, l42: ", with a mole fraction range of 350-520 parts per trillion (ppt) in the unpolluted free troposphere (Berry 2013, Remaud 2023)."

*AR: added.*

P2, l47: "to CO2 and sulfur dioxide (SO2), a precursor..."

*AR: added.*

P5, l142: What is the COS mole fraction in that gas?

*AR: this air mixture was made in the laboratory and contained 0 $N_2O$ and 0 COS. It has now been specified in the text: "[...] with air from a cylinder of synthetic air mixed with low*

mole fractions of CH$_4$, CO$_2$ and CO, which was meant to simulate stratospheric air conditions *and contained 0 ppb N$_2$O and 0 ppt COS."*

P6, l 169: Are these valves used for heating?

*AR: yes, as a coincidental feature the activation of these valves causes them to heat up, as the activation is triggered by an electrical coil. This coil creates the magnetic field to operate the valve and generates heat as a byproduct of its operation.*

P6, l181: Is the number of 1800 s correct, i.e., 30 minutes?

*AR: yes. BigLISA flew on a gondola attached to a large balloon. The gondola underwent slow descent following the balloon's burst. This allowed for longer sample collection times, particularly for the highest-altitude samples.*

P9, l217: Should this be section number 2.3.1 SPIRALE, instead of 2.4.1 SPIRALE?

*AR: thank you for noting this. It has now been corrected.*

P10, l223: Should this be section number 2.3.2 ACE-FTS, instead of 2.4.2 ACE-FTS?

*AR: corrected.*

P12, l272: Can you provide details of the differential pressure sensors? Could be quite useful to know for interested audience, in terms of materials to avoid.

*AR: the specific brand and model have been included (now Line 293):* **Amsys, model AMS5915_0050_D_B**.

P14, l327: Is the "self-consistent" statement fully applicable when comparing systems that sample either during ascend or during descend?

*AR: thank you for raising the good question. For regular weather balloon flights, the ascent and descent phases take a similar amount of time. Furthermore, the payload does not travel a significant horizontal distance during the flight, which is particularly true for the stratospheric sampling. Therefore, results from both phases should not differ greatly. For clarity, "self-consistent" was changed to "consistent".*

P14, l331: I'd suggest making the statement in that sentence more clear and spell out the conclusion with more clarity, for example: "However, we assume that differences due to instrumental effects remain marginal, while we believe that the day-to-day variability and long-term trends in COS mole fractions are the most important cause for the observed differences."

*AR: thank you for this suggestion, and we have changed the sentence to **"Therefore, we assume that differences due to instrumental effects remain marginal, while we believe that***

*the day-to-day variability and long-term trends in COS mole fractions are the most important cause for the observed differences."*

P14, Table caption: Should this be expanded a bit? It is not clear to me what this exactly means. Are there data gaps at specific heights, and are these suggested to be due to contamination from glue, surface effects, and differential pressure sensors? If so, how could this only affect short sections/a small number of the sample, but not all samples?

*AR: the caption has been changed to "**data gaps caused by clear contamination effects and their respective causes in different AirCore profiles**". During both sampling and analysis, air flows into and out of the AirCore in nearly a plug flow, causing very little mixing between adjacent portions of sampled air (Karion et al., 2010; Membrive et al., 2017; Tans, 2022). Therefore, if a contamination source is present along the coil, the contamination is likely to show as a "localised" feature rather than spreading through all the sample.*

P15, l334: Maybe add "as" to "and as we have no O3 measurements"?

*AR: done.*

P15, l358: Can you be more specific on "the variability" you refer to? Could you please explain what difference have in mind here?

*AR: "variability" was changed to "[...] in spite of the observed **differences between the retrieved profiles**, all profiles [...]".*

P17, l386: Example where a marker/arrow within the data figure to identify/highlight that event could be helpful to better understand what the discussion is about.

*AR: we recognise that adding markers or arrows may generally help the reader in identifying specific features in a figure. However, the authors believe that in this case the introduction of an arrow may collaterally cover other data, moving the focus away from the main theme of the discussion.*

P20, Figure captions: Here and in other figures where relevant, can the time interval represented by the modelled COS profiles be included into the caption as well?

*AR: the time interval over which the remote sensing COS observations of ACE-FTS were averaged has been included in the captions.*

P22, Figure 6: Here and in other figures where relevant, can the regression functions be included into the figures? This would be much easier to follow than having them in a separate table. If they are needed in a table, I suggest showing in both forms.

*AR: the authors would prefer to exclude additional text from figures that are already rather busy. Tables have now been moved to be adjacent to their respective figures – the authors*

*believe this solution to be clearer and more informative, since in the Tables it is also possible to report the uncertainties related to the results of each regression.*

P24, l470: Say which laboratory.

**AR:** *"[...] with a QCLS in* **the CIO laboratory at the University of Groningen** *[...]"*

P25, l480: Change to "different sorts of inlets with different O3 scrubber materials."

**AR:** *done.*

P25, l482: "… variability, a hypothesis…"

**AR:** *done.*

Supplements, Figures 7-15:

- Where indicated in the figure legend, I am unable to tell ascent data and descent data apart; all I see is one colour.

**AR:** *the plots were checked to be colour blindness proof. For the TRN flights, Fig. S7-S9 (now Fig. S5-S7) ascent data was not available. The figures have been corrected.*

- Include in captions that the tropopause height is indicated by the green line.

**AR:** *done.*